# DeepRepoQA: Code Repository Question Answering with Deep Agent Exploration

## Abstract

Answering developer questions about a software repository is a critical yet under-explored problem in software engineering. While existing repository understanding methods have advanced the field, they predominantly rely on surface-level code retrieval and lack the ability for deep reasoning over multiple files, complex software architectures, and grounding answers in long-range code dependencies. To address these limitations, we propose DeepRepoQA, a novel question answering (QA) framework for repository-level code understanding. DeepRepoQA builds on the agentic framework where LLM agents find answers through a systematic tree search over the repository structure. A Monte-Carlo Tree Search (MCTS) mechanism is employed to empower agents to dynamically search, navigate, and inspect code dependencies, enabling effective multi-hop reasoning over long-range code dependencies. Comprehensive experiments on the SWE-QA benchmark demonstrate substantial performance gains over strong baselines, validating the effectiveness of systematic MCTS-guided exploration for multi-hop repository reasoning[1].

## 1 Introduction

Answering questions about a software repository is an underexplored yet critical problem (Liu & Wan, 2021; Peng et al., 2025; Chen et al., 2025a). To accomplish a function or resolve an issue, developers frequently need to navigate large, interconnected codebases, trace dependencies across multiple files, and synthesize architectural knowledge in order to answer complex questions (Jimenez et al., 2024; Zhang et al., 2023; Ouyang et al., 2024). With the rapid advancement of LLMs (Yao et al., 2023; Yang et al., 2024), there is growing interest in repository-aware assistants that can provide end-to-end support for such tasks.

Despite rapid progress, existing repository understanding methods often operate at the function level (Huang et al., 2021; Liu & Wan, 2021; Lee et al., 2022; Li et al., 2024; Sahu et al., 2024), or rely on basic retrieval-augmented generation (RAG) that surfaces local fragments of code (Zhang et al., 2023). These approaches are ill-suited to deep repository reasoning where answers require following call chains across files, combining dispersed evidence, and articulating architectural intent (Ouyang et al., 2024). In particular, flat RAG systems provide little control over search, struggle to balance exploration and exploitation, and frequently suffer from context loss or short-sighted retrieval loops.

To address these limitations, we propose DeepRepoQA, a novel agentic question answering (QA) framework for deep repository understanding. Unlike flat retrieval pipelines (Zhang et al., 2023), DeepRepoQA casts QA as a deep search-and-verify process over the repository structure, enabling LLM agents to systematically explore repositories and synthesize evidence before producing answers. The framework defines a compact, semantically meaningful action space spanning semantic search, structural navigation, and targeted code inspection. Given a question, the agent initializes a search tree and iteratively (i) perceives the current state and reasoning trajectory, (ii) plans the next step, (iii) executes the action over an index-backed representation to gather verifiable evidence, and (iv) evaluates utility to steer future exploration. Collected evidence is curated and appended to a memory path from the root to the current node; this path structures prompts, stabilizes reasoning, and, once

---

[1] Our code and data are publicly available at `https://anonymous.4open.science/r/DeepRepoQA-DAC4`

sufficient evidence is amassed, enables synthesis of a final answer grounded in cited code spans. Compared to single-path ReAct-style agents (Peng et al., 2025), our approach introduces two key innovations: (1) an iterative search-and-verify process via MCTS for multi-hop repository reasoning, and (2) learned priors/values from LLM feedback that reduce search depth and reduce drift. Together, these elements transform repository QA from ad-hoc retrieval to systematic search with consistent, evidence-based answers.

We evaluate DEEPREPOQA on SWE-QA benchmark (Peng et al., 2025) across multiple LLM backbones such as GPT-4o and Claude 3.7 Sonnet. DEEPREPOQA demonstrates consistent gains over strong baselines. The largest overall improvement over SWE-QA-AGENT exceeds 11.9% on GPT-4o. DEEPREPOQA also surpasses leading commercial tools such as Cursor and Tongyi Lingma with DeepSeek-V3.1 and Claude 3.7 Sonnet. Ablations confirm that MCTS is the most critical component, while semantic search, the evaluation, and perception agents further improve stability and efficiency. Furthermore, a case study demonstrates that our MCTS-guided, verification-centric exploration mitigates retrieval bias by grounding answers in cross-file evidence.

Our contributions are as follows:

- We propose an agentic QA framework for repository understanding, enabling structured, verifiable multi-hop reasoning over long-range code dependencies.
- We introduce MCTS-guided exploration with LLM feedback and memory for efficient, repository-grounded search.
- We conduct a comprehensive evaluation on SWE-QA, demonstrating consistent improvements over state-of-the-art baselines.

## 2 RELATED WORK

Repository-level code understanding has become a central challenge for modern software development, such as automated code generation (Shrivastava et al., 2023; Zhang et al., 2024; Shi et al., 2024), code translation (Wang et al., 2024a; 2025a), and issue resolution (Jimenez et al., 2024; Chen et al., 2025b). These approaches include retrieval-augmented context gathering (Zhang et al., 2023), agent-based repository navigation (Ma et al., 2024; Yang et al., 2024; Li et al., 2025), graph-based cross-file relation modeling (Ouyang et al., 2024), and context-aware code completion (Shrivastava et al., 2023). While highly effective for their intended purposes, they primarily focus on code synthesis and modification, rather than on the comprehensive, multi-hop reasoning required for deep question answering.

In code question answering, most existing work targets snippet or function-level understanding. Studies include neural QA systems for subroutine behaviors (Bansal et al., 2021), task-adaptive pre-training for code QA (Yu et al., 2022), and transformer-based models for regulatory codes (Xue et al., 2024). Recently, attention has shifted to repository-level QA, which addresses architectural questions requiring multi-file, multi-hop reasoning across entire codebases. While benchmarks and improvements have been proposed (Chen et al., 2025a; Strich et al., 2024; Peng et al., 2025), the field remains nascent. Analyses indicate that directly applying LLMs and RAG to repository-level QA faces inherent limitations (Andryushchenko et al., 2024), highlighting the need for more structured reasoning approaches.

Comparatively, our DEEPREPOQA overcomes these limitations by formulating repository QA as a planning problem. Using a multi-module Monte Carlo Tree Search (MCTS), our framework enables a principled and systematic exploration of the vast and complex reasoning space within a codebase. This supports deliberate, multi-hop inference across entire repositories, surpassing simple retrieval or localized analysis.

## 3 APPROACH

We propose DEEPREPOQA, a repository QA method that leverages deep agent exploration. DEEP-REPOQA adopts an agentic framework grounded in Monte Carlo Tree Search (MCTS), where four specialized agents, responsible for perception, planning, execution, and evaluation, collaborate to enable deep exploration of the codebase. As illustrated in Figure 1, DEEPREPOQA answers

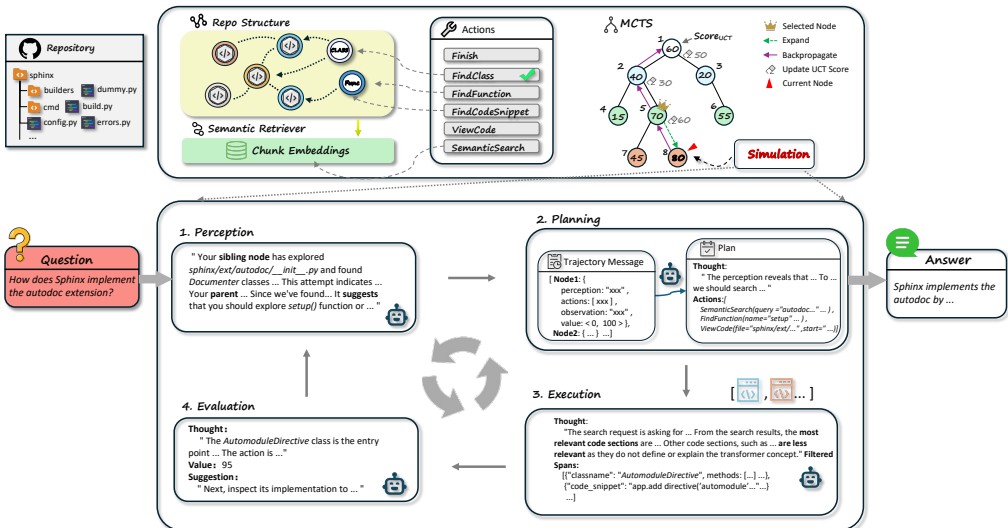

Figure 1: Overview of the DEEPREPOQA framework. The system answers repository-level questions through an iterative search-and-verify loop guided by an MCTS process. In each iteration, it perceives the current state, plans the next action, executes it to collect code evidence, and evaluates the outcome to steer subsequent exploration. Once sufficient evidence is gathered, the system synthesizes a final answer grounded in explicit code citations.

repository questions through an iterative search-and-verify loop. Upon receiving a question, the system constructs a compact search tree representing potential next steps (e.g., search, navigate, inspect, or stop). Each step in this loop consists of four stages governed by the corresponding agents: a *Perception Agent* analyzes the current state of exploration. A *Planning Agent* then proposes promising subsequent actions based on the perceived context. An *Execution Agent* performs these actions to gather concrete evidence from the code. Finally, an *Evaluation Agent* assesses the results, assigns value scores, and propagates feedback to guide subsequent exploration. This cycle repeats until sufficient evidence has been accumulated, after which the system synthesizes a final answer grounded in explicit code citations. Through this structured, feedback-driven process, DEEPREPOQA transforms repository QA from a linear retrieval task into a multi-path reasoning process, thereby enabling effective handling of complex, multi-hop questions. Detailed descriptions of each agent are provided in the following sections.

## 3.1 REPOSITORY PARSING AND REPRESENTATION

DeepRepoQA's ability to comprehend and reason over large codebases relies on a systematic *repository parsing* and *representation* pipeline. The goal of this pipeline is to transform the raw repository into structured information that can be efficiently queried and analyzed by the model.

**Repository Parsing** The parsing process involves traversing the files and directory structure of the repository to identify and extract relevant source code elements. Specifically, DeepRepoQA analyzes files to detect classes, functions, and code snippets, capturing their essential syntactic and semantic information. This process leverages language-specific parsers and heuristics based on common project layouts (e.g., `src`, `tests`), and employs Tree-sitter[2] to obtain lightweight, language-agnostic Abstract Syntax Trees that support precise structural extraction. Parsing also includes resolving cross-file relationships, such as function calls, class inheritance, and module dependencies, enabling the agent to build a comprehensive view of the repository's overall architecture.

---

[2]https://tree-sitter.github.io/tree-sitter/

**Repository Representation** Once parsing is complete, the extracted information is transformed into structured representations suitable for downstream reasoning. DeepRepoQA employs a multi-level representation strategy:

- **Index-based Lookup:** The repository directory structure and AST trees are modeled to help the agent understand the positions and contexts of classes and functions within modules. By directly parsing ASTs, the agent can precisely locate classes, functions, and code snippets without relying on fuzzy or semantic search, enabling structure-aware retrieval via `FindClass`, `FindFunction`, and `FindCodeSnippet`.

- **Semantic Retriever:** Code elements requiring conceptual-level retrieval (e.g., arbitrary code snippets) are vectorized to support semantic search and reasoning. Indexed nodes are embedded and queried using a RAG-style Semantic Retriever, corresponding to the `SemanticSearch` operation in the system.

By combining these representations, DeepRepoQA achieves both *high-level structural understanding* and *fine-grained semantic comprehension*, enabling effective repository analysis and intelligent question-answering.

### 3.2 ACTION SPACE

The DeepRepoQA agent operates with a discrete action space consisting of six core capabilities. These actions equip the agent with essential tools for repository comprehension, targeted code retrieval, and answer synthesis. The available actions are:

- **FindClass**: enables the agent to locate class definitions within the repository. This action supports high-level structural reasoning by identifying key abstractions, inheritance relationships, and encapsulated functionalities. It is particularly useful for understanding object-oriented designs and module interactions.

- **FindFunction**: allows the agent to identify specific function or method definitions. This facilitates fine-grained understanding of code behavior, enabling the agent to trace logic, inputs, and outputs within targeted modules.

- **FindCodeSnippet**: provides the agent with the ability to extract relevant code fragments based on contextual needs. This action is essential for gathering actionable examples, verifying implementation details, or focusing on critical sections within large files.

- **SemanticSearch**: empowers the agent to perform advanced search over repository content using semantic embeddings rather than mere keyword matching. Leveraging a code embedding model, this action allows the agent to retrieve conceptually related code or documentation, improving recall across complex codebases.

- **ViewCode**: grants the agent the ability to inspect the full contents of specific files. This low-level inspection is useful for understanding implementation details, configuration, and documentation. Standard command-line utilities such as cat or grep may be executed to facilitate precise content access.

- **Finish**: marks the completion of the answer task, indicating that the agent has synthesized a definitive answer from the collected information or reaches a predefined maximum number of iterations.

### 3.3 AGENTS

**Perception Agent** The *Perception Agent* analyzes the *current search state* before any new action is proposed, serving as the sensory input for the MCTS framework. It synthesizes the trajectory from the root to the current node, along with information from sibling branches, to produce a concise situational report. This report serves multiple purposes: it detects redundancy to avoid re-exploring dead ends, highlights unvisited but promising code regions, surfaces conflicts or gaps in the collected evidence, and extracts structured cues (e.g., candidate symbols, files, or API calls) for further investigation. For a question like, "How does Sphinx implement the autodoc extension?", if one branch has explored `sphinx/ext/autodoc/__init__.py` and found `Documenter` classes, the Perception Agent

may notice that the critical `setup()` function, the standard entry point for Sphinx extensions, has not been analyzed. It will therefore flag the `setup()` function as a high-priority target to guide the next planning phase toward discovering how the extension integrates with Sphinx's core. By delivering a compact, noise-reduced state assessment to downstream agents, it ensures that subsequent decisions are based on a holistic view of the search progress and blind spots.

**Planning Agent** The *Planning Agent* is the core decision-maker, responsible for turning the situational report from the Perception Agent into a concrete action plan. When a leaf node is selected for expansion, this agent generates a set of candidate actions to create new child nodes, effectively growing the search frontier. It inspects the curated state summary to propose diverse and strategic next steps, while explicitly avoiding redundant trials already attempted in sibling branches. Building on the same example in which perception report flagged the `setup()` function as a key target, the Planning Agent might generate three candidate actions for expansion: 1) a `FindFunction` call for `setup` within the `sphinx/ext/autodoc/` directory, 2) a `SemanticSearch` for "autodoc directive registration" to find where the `automodule` directive is defined, and 3) a `ViewCode` action on `sphinx/ext/autodoc/__init__.py` to manually locate the `setup` function. Each proposed action becomes a new child node in the search tree, representing a distinct reasoning path conditioned on both long-term goals and expected immediate information gain. The planning prompt is provided in Appendix A.6.

**Execution Agent** The *Execution Agent* leverages the repository representations and action space introduced in Section 3.1and 3.2 to carry out the planner-selected actions over the unified structural–semantic repository index. It produces raw outputs that include results from structure-aware lookup, semantic retrieval, and file-level inspection. Fuzzy name matching is applied to mitigate minor discrepancies in model-generated queries. Immediately after execution, the evidence-curation stage filters and normalizes the raw outputs: it ranks evidence by task relevance, de-duplicates overlapping spans, collapses boilerplate, and extracts focused quotations with exact file and line ranges. Low-value or noisy snippets are discarded, and salient context is stitched into a compact, citation-ready bundle. This curated evidence set becomes the observation for the current node and the context for subsequent reasoning. The prompt used by the filtering component is provided in Appendix 3. Detailed argument definitions for all actions are listed in Appendix A.2.

**Evaluation Agent** The *Evaluation Agent* assesses the outcome of an executed action, providing critical feedback for the MCTS learning loop. After the Execution Agent returns a curated piece of evidence, this agent scores the action-observation pair by estimating a scalar value representing its utility in answering the original question. It also generates targeted, qualitative suggestions for subsequent steps, such as "verify a dependency path" or "inspect call sites." For instance, if the `FindFunction` action for `setup` returns a code snippet containing the call `app.add_directive('automodule', AutomoduleDirective)`, the Evaluation Agent would assign a high value (e.g., 95) to this node, as it directly reveals the link between the directive name and its implementing class. Its feedback might suggest, "The `AutomoduleDirective` class is the entry point for the directive. Next, inspect its implementation to understand how it processes the directive's content." This value is then back-propagated up the search tree, updating the visit counts and value estimates of all nodes on the path from the current node to the root. This feedback loop allows the agent to continuously refine its search policy, prioritizing more promising reasoning paths. The evaluation prompt is provided in Appendix A.6.

### 3.4 Monte-Carlo Tree Search Process

The four agents collaborate within the MCTS framework to answer a user's question. Starting from the question, DEEPREPOQA builds a small search tree and repeatedly runs the four-agent simulation: the *Perception Agent* forms a situational report, the *Planning Agent* proposes an action, the *Execution Agent* performs it with internal *Filtering* to curate evidence, and the *Evaluation Agent* scores utility and guidance. MCTS selects and expands promising branches until `Finish` is chosen, after which we synthesize a concise, citation-backed answer. The full pseudocode is provided in Appendix A.3. The MCTS engine guides repository exploration through four phases—selection, expansion, simulation, and back-propagation—designed to systematically navigate the complex search space of repository-level reasoning[3].

---

[3]The workflow of our MCTS is illustrated in Figure 5 (Appendix A.4).

**Selection**  Starting from the root node, child nodes are selected recursively according to the Upper Confidence Bound (UCT) policy until reaching a leaf node. The selection strategy balances exploration and exploitation using the UCT formula:

$$\text{UCT}(s, a) = Q(s, a) + c \cdot \sqrt{\frac{\ln N(s)}{N(s, a)}} \tag{1}$$

where $Q(s, a)$ is the average value of taking action $a$ at state $s$, $N(s)$ is the visit count of $s$, $N(s, a)$ is the visit count of $(s, a)$, and $c$ is the exploration coefficient.

**Expansion**  When selection reaches a node that is not fully expanded, a new child node is added. Before proposing candidates, the *Perception Agent* produces a concise situational report from the direct trajectory and sibling attempts, highlighting redundancy, gaps, and promising regions. This lateral awareness lets planning avoid repeated trials without storing full sibling transcripts, since the perception report summarizes the relevant horizontal context.

**Simulation**  Unlike traditional MCTS rollouts, our simulation is a single-step agent loop: *Perception* summarizes state and siblings, *Planning* proposes the next action, *Execution* runs it and performs internal *Filtering* to curate evidence, and *Evaluation* assigns a scalar value. The evaluation output $V(s, a)$ directly sets $Q(s, a) \leftarrow V(s, a)$, allowing effective assessment without multi-step rollouts and reducing compute while preserving decision quality.

**Back-propagation**  The simulation value is back-propagated to update $Q$ and visit counts along the path to the root. High-quality nodes are revisited more often, while low-value branches are implicitly pruned as their selection probability drops, focusing exploration on promising regions over time.

**Answer Synthesis and Termination**  When the `Finish` action is selected, the system composes the final answer by summarizing the reasoning trace from the best trajectory and citing decisive evidence with file and line spans for traceability. We require at least one supporting code span to ensure grounding. The process concludes by validating that the answer addresses the question type ("what," "where," "how," or "why").

## 4 EXPERIMENTS

We evaluate the effectiveness of DEEPREPOQA across multiple dimensions, aiming to answer the following research questions:

- **RQ1**: How effectively does DEEPREPOQA answer code questions at repository scale?
- **RQ2**: To what extent do individual components influence the overall performance of our method?
- **RQ3**: How does the number of exploration iterations in DEEPREPOQA affect the answering performance?

### 4.1 EXPERIMENTAL SETUP

**Evaluated Methods**  We evaluate our DEEPREPOQA against the following baselines. **Direct Prompting** involves querying the LLMs directly without providing any repository context, testing their intrinsic knowledge. For **RAG-based Methods**, we evaluate two retrieval-augmented generation strategies: **Function Chunking RAG** (Wang et al., 2024b), which parses the repository into function-level chunks to create a semantic code index for retrieval, and **Sliding Window RAG** (Zhang et al., 2023), which uses a sliding window to segment code files into overlapping chunks for retrieval. For **Agent-based Methods**, we compare against **SWE-QA-AGENT** (Peng et al., 2025), a ReAct-style agent designed for repository-level QA that uses a set of tools for iterative reasoning and information retrieval. Finally, we also compare against two state-of-the-art **Commercial Tools**, **Tongyi Lingma** (VSCode plugin v2.5.16, auto mode) and **Cursor** (v2025.09.04-fc40cd1, auto mode), to benchmark against proprietary systems.

All open-source methods are evaluated with the same set of underlying LLMs to ensure a fair comparison, including three powerful and representative models: DeepSeek-V3.1 (DeepSeek-AI, 2025), GPT-4o (OpenAI, 2024), and Claude 3.7 Sonnet (Anthropic, 2025). For the commercial

Table 1: Overall results on SWE-QA. DEEPREPOQA yields the highest overall score across all backbones. Gains concentrate on Completeness and Reasoning, reflecting deeper multi-hop grounding rather than surface retrieval.

| Methods | Evaluation Metrics | | | | | Overall |
|---|---|---|---|---|---|---|
| | Correctness | Completeness | Relevance | Clarity | Reasoning | |
| *Commercial Tools* | | | | | | |
| Tongyi Lingma | 8.15 | 6.58 | 9.87 | 9.47 | 8.31 | 42.38 |
| Cursor | 8.46 | 7.55 | 9.85 | 9.51 | 8.68 | 44.05 |
| *GPT-4o* | | | | | | |
| Direct Prompting | 6.48 | 3.19 | 9.30 | **9.66** | 4.88 | 33.50 |
| Function Chunking RAG | 6.83 (+0.35) | 4.02 (+0.83) | 9.29 (-0.01) | 9.38 (-0.28) | 6.31 (+1.43) | 35.83 (+2.33) |
| Sliding Window RAG | 6.80 (+0.32) | 3.96 (+0.77) | 9.30 (+0.00) | 9.38 (-0.28) | 6.33 (+1.45) | 35.77 (+2.27) |
| SWE-QA-Agent | 6.79 (+0.31) | 5.03 (+1.84) | 9.05 (-0.25) | 9.06 (-0.60) | 7.26 (+2.38) | 37.18 (+3.68) |
| DEEPREPOQA | **7.87** (+1.39) | **6.66** (+3.47) | **9.59** (+0.29) | 9.28 (-0.38) | **8.22** (+3.34) | **41.61** (+8.11) |
| *DeepSeek V3.1* | | | | | | |
| Direct Prompting | 6.42 | 3.50 | 9.09 | 9.45 | 5.24 | 33.71 |
| Function Chunking RAG | 6.92 (+0.50) | 4.61 (+1.11) | 9.60 (+0.51) | 8.92 (-0.53) | 6.78 (+1.54) | 36.83 (+3.12) |
| Sliding Window RAG | 6.57 (+0.15) | 4.37 (+0.87) | 9.62 (+0.53) | 8.98 (-0.47) | 6.51 (+1.27) | 36.04 (+2.33) |
| SWE-QA-Agent | 8.40 (+1.98) | 7.16 (+3.66) | 9.62 (+0.53) | 9.28 (-0.16) | 8.36 (+3.12) | 42.70 (+8.99) |
| DEEPREPOQA | **8.57** (+2.15) | **8.04** (+4.55) | **9.83** (+0.74) | **9.29** (-0.17) | **8.89** (+3.64) | **44.62** (+10.91) |
| *Claude 3.7 Sonnet* | | | | | | |
| Direct Prompting | 6.79 | 4.06 | 9.27 | 9.47 | 6.32 | 35.91 |
| Function Chunking RAG | 7.95 (+1.17) | 6.72 (+2.66) | 9.68 (+0.41) | 9.28 (-0.19) | 8.42 (+2.10) | 42.07 (+6.16) |
| Sliding Window RAG | 7.93 (+1.14) | 6.72 (+2.66) | 9.65 (+0.38) | 9.27 (-0.19) | 8.38 (+2.06) | 41.96 (+6.06) |
| SWE-QA-Agent | 8.37 (+1.59) | 7.58 (+3.51) | 9.83 (+0.56) | 9.36 (-0.11) | 8.71 (+2.39) | 43.84 (+7.94) |
| DEEPREPOQA | **8.90** (+2.12) | **8.64** (+4.58) | **9.91** (+0.64) | **9.48** (+0.01) | **9.15** (+2.83) | **46.08** (+10.17) |

tools, Tongyi Lingma uses its proprietary model, and Cursor runs in its default "auto" mode that automatically selects the best model based on the user query with built-in retrieval and orchestration.

**Dataset** We evaluate all methods on the SWE-QA benchmark (Peng et al., 2025). SWE-QA is a repository-level code question answering benchmark designed to evaluate automated QA systems in realistic software development environments. The benchmark consists of 576 high-quality question-answer pairs from 12 diverse and popular open-source Python repositories.

**Metrics** We adopt the LLM-as-a-Judge evaluation protocol from (Peng et al., 2025) to assess answer quality. This approach demonstrates high reliability and strong correlation with human judgments in both natural language generation and software engineering tasks (Liu et al., 2023; Wang et al., 2025b). Following this protocol, we employ GPT-5 (OpenAI, 2025) as the judge. The evaluation is conducted across five key dimensions: (1) **Correctness**, which assesses the factual accuracy of the answer; (2) **Completeness**, evaluating how thoroughly the answer addresses all aspects of the question; (3) **Relevance**, about how well the answer matches the query; (4) **Clarity**, about readability and ease of understanding; and (5) **Reasoning**, which examines the logical coherence and strength of the argument. Each dimension is rated on a scale from 0 to 10. To ensure a reliable and unbiased evaluation, the judge model is different from the models being evaluated, and the order of candidate answers is randomized. The prompt used by the LLM-as-a-Judge is provided in Appendix A.6.

**Implementation Details** Our `SemanticSearch` component employs the `voyage-code-3` code embedding model. For MCTS, we cap the number of children expanded per node at three (*max_expand* = 3) and set the maximum exploration iterations to 20 for the main results reported in Table 1. For answer generation across all evaluated backbones, we use each provider's API default decoding parameters (e.g., temperature, top_p) without additional tuning.

### 4.2 MAIN RESULTS (RQ1)

Table 1 compares the overall results by various methods. Across models, DEEPREPOQA is the strongest open-source system and closes the gap to commercial tools. With DeepSeek-V3.1 it achieves an overall score of 44.62, surpassing Cursor at 44.05 and Tongyi Lingma at 42.38. With

Table 2: Ablations on key components. Removing MCTS causes the largest drop, confirming exploration–exploitation control is critical; perception/evaluation agents and semantic search further stabilize evidence curation and improve efficiency.

| Variant | Evaluation Metrics | | | | | Overall |
|---|---|---|---|---|---|---|
| | Correctness | Completeness | Relevance | Clarity | Reasoning | |
| DEEPREPOQA | 8.57 | 8.04 | 9.83 | 9.29 | 8.88 | 44.62 |
| w/o MCTS | 8.18 (-0.39) | 7.62 (-0.42) | 9.27 (-0.56) | 8.88 (-0.41) | 8.39 (-0.49) | 42.34 (-2.28) |
| w/o Perception Agent | 8.37 (-0.20) | 7.89 (-0.15) | 9.63 (-0.20) | 9.19 (-0.10) | 8.67 (-0.21) | 43.75 (-0.87) |
| w/o Evaluation Agent | 8.21 (-0.36) | 7.52 (-0.52) | 9.62 (-0.21) | 9.16 (-0.13) | 8.67 (-0.21) | 43.18 (-1.44) |
| w/o Semantic Search | 8.28 (-0.29) | 7.81 (-0.23) | 9.49 (-0.34) | 9.00 (-0.29) | 8.57 (-0.31) | 43.15 (-1.47) |

GPT-4o and Claude 3.7 Sonnet it remains the top open-source method; on Claude it also exceeds commercial systems, while on GPT-4o it narrows the remaining gap.

Gains concentrate on Completeness and Reasoning—the dimensions most sensitive to multi-hop retrieval and evidence synthesis—while maintaining strong Correctness and Relevance and fluent Clarity. This consistent pattern indicates that MCTS-guided exploration and index-grounded actions yield more thorough and coherent answers.

RAG variants outperform direct prompting but still trail agentic approaches, particularly on Completeness. DEEPREPOQA strengthens the substantive dimensions across backbones and, with stronger backbones, reaches or exceeds commercial performance. Overall, systematic, feedback-guided search over repositories proves more effective than flat retrieval or single-trajectory agents.

## 4.3 ABLATION STUDY (RQ2)

To understand the contribution of individual components, we conduct ablation studies removing: (i) MCTS framework (replaced with greedy single-trajectory search), (ii) Perception Agent (without state summarization), (iii) Evaluation Agent (reverted to rollout-based evaluation), and (iv) Semantic Search (exact name-based retrieval only). Results in Table 2 show MCTS as the most critical component, with its removal causing the largest performance drop of 2.28 points (5.1% relative decline), followed by Semantic Search (-1.47 points), Evaluation Agent (-1.44 points), and Perception Agent (-0.87 points). The MCTS framework's substantial impact across all evaluation dimensions confirms that systematic exploration-exploitation balance is fundamental for repository-level reasoning.

The component hierarchy reveals that systematic exploration mechanisms outweigh individual optimization techniques in importance. The notable impact of removing the Evaluation Agent validates our design choice to replace expensive rollouts with learned value estimation, while semantic search contributes balanced improvements across diverse question types. These findings demonstrate that principled algorithmic innovations (MCTS, value estimation, semantic actions) drive primary performance gains, providing clear guidance for future development priorities in repository-level QA systems.

## 4.4 IMPACT OF EXPLORATION ITERATIONS (RQ3)

To address RQ3, we analyze the impact of varying the maximum number of exploration iterations on the performance of DEEPREPOQA. The results with DeepSeek-V3.1 as the base model are summarized in Table 3. We can observe that increasing the maximum number of explored nodes steadily improves performance: the overall score rises from 43.06 (3 nodes) to 44.50 (10 nodes, +1.44) and 44.62 (20 nodes, +0.12). Correctness increases from 8.27 to 8.57 (+0.30), Relevance from 9.71 to 9.83 (+0.12), and Reasoning from 8.81 to 8.88 (+0.07), while Completeness peaks at 8.10 around 10 nodes and slightly tapers to 8.04 at 20. In practice, ten nodes offer a strong quality–latency trade-off, whereas twenty nodes maximize answer quality with diminishing returns.

We further analyze the agent's exploration behavior by examining its iteration depth and action usage. The results are summarized in Figure 2 and Figure 3. The agent's exploration behavior offers further insight into these performance gains. Analysis of exploration depth (Figure 2) reveals distinct strategies across models, which correlate strongly with the final scores in Table 1. For

Table 3: Effect of exploration budget. Increasing the max explored nodes steadily improves quality with favorable latency trade-offs; moderate budgets already capture most gains by enabling verification of cross-file evidence.

| Max Nodes | Evaluation Metrics | | | | | Overall |
|---|---|---|---|---|---|---|
| | Correctness | Completeness | Relevance | Clarity | Reasoning | |
| 3 | 8.27 | 7.97 | 9.71 | 9.30 | 8.81 | 43.06 |
| 5 | 8.43 | 7.99 | 9.64 | 9.21 | 8.70 | 43.97 |
| 10 | 8.47 | 8.10 | 9.79 | 9.29 | 8.86 | 44.50 |
| 20 | 8.57 | 8.04 | 9.83 | 9.29 | 8.88 | 44.62 |

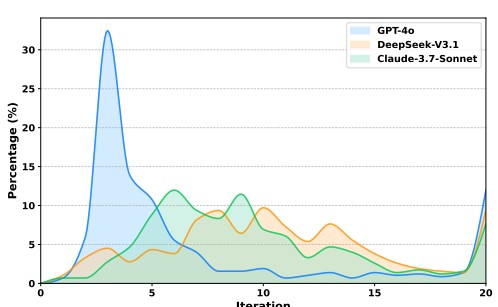

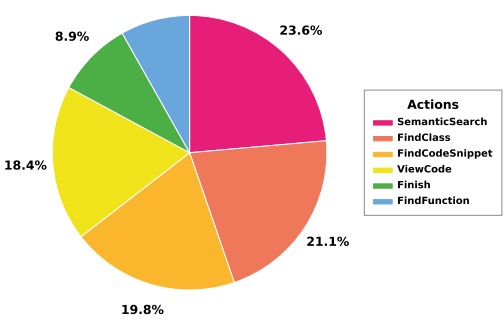

Figure 2: Distribution of exploration iterations by backbone. DeepSeek-V3.1 and Claude-3.7-Sonnet typically explore deeper with more actions, enabling broader evidence gathering and higher answer quality.

Figure 3: Action usage during trajectories. Trajectories are dominated by `SemanticSearch` (broad hypotheses) and structure-aware navigation (`FindClass`, `FindCodeSnippet`), with frequent `ViewCode` for verification—evidence of a search-and-verify loop.

instance, GPT-4o often terminates with a shallow search of 3–5 iterations, corresponding to lower Completeness and Reasoning scores and an overall score of 41.61. In contrast, Claude 3.7 Sonnet and DeepSeek-V3.1 favor medium-to-deep exploration, achieving higher overall scores of 46.08 and 44.62, respectively. While returns are not strictly monotonic—as excessive search can introduce latency with marginal benefits—a moderate search depth systematically improves evidence coverage and synthesis. For DeepSeek-V3.1, the performance gains are most pronounced in the medium-depth regime of around 10 nodes, capturing the majority of the improvement.

Beyond iteration depth, the distribution of actions used during trajectories (Figure 3) clarifies the underlying mechanism. The agent's behavior is dominated by a combination of broad, meaning-based exploration (`SemanticSearch`) and targeted, structure-aware navigation (`FindClass`, `FindCodeSnippet`), supplemented by frequent use of `ViewCode` for verification. Actions like `FindFunction` and the terminal `Finish` action are less common, indicating that the agent spends most of its budget actively gathering and confirming evidence. This pattern of guided exploration followed by verification directly contributes to the observed improvements in Completeness and Reasoning as the exploration budget increases, as it allows the agent to build a more comprehensive and well-grounded understanding of the repository before synthesizing an answer.

## 4.5 CASE STUDY

To illustrate how different methods perform in terms of answer quality, we examine a question from scikit-learn: "*What does sklearn.preprocessing.StandardScaler do to target variables?*" As shown in Figure 4, the baseline **SWE-QA-Agent** mistakenly "assumes" the target vector should be treated as a feature vector. This error stems from the fact that the retrieved code snippets almost entirely concern how `StandardScaler` processes feature vectors. As a result, the answer naturally focuses on explaining the standardization of features, while overlooking the design of `StandardScaler` regarding target vectors—it does not process target vectors at all and simply ignores the y parameter in the fit method. In contrast, our **DEEPREPOQA** can deeply understand `StandardScaler`'s API design and retrieve crucial contextual information (such as the existence

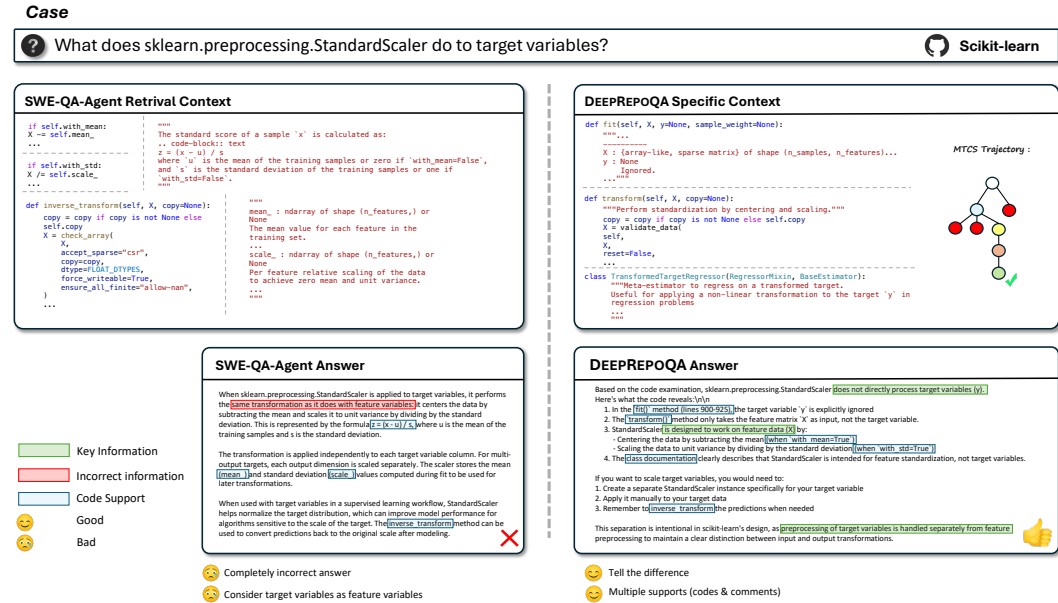

Figure 4: Case study: scikit-learn `StandardScaler`. The baseline SWE-QA-AGENT conflates targets with features and claims targets are standardized; DEEPREPOQA follows code-grounded evidence to show `StandardScaler` ignores `y` ("y: None, Ignored" in `fit`) and only transforms `X`, and identifies `TransformedTargetRegressor` as the correct mechanism for target scaling.

of `TransformedTargetRegressor`), enabling it to accurately explain the architectural design principles and how to properly standardize target variables. Starting from `StandardScaler`'s basic usage, our agent infers potential misconceptions about target variable processing, then locates the fit method's parameter documentation in the source code, clearly showing "y: None, Ignored"; it then identifies the `transform` method's design that only accepts `X` parameter; and finally verifies sklearn's provision of `TransformedTargetRegressor` specifically for target variable transformation. These corroborating code fragments jointly ground the final answer.

## 5 CONCLUSION

In this paper, we introduced DEEPREPOQA, a novel agent-based framework that reformulates repository-level code question answering as a planning problem solved with Monte Carlo Tree Search. Comprehensive evaluations demonstrate that DEEPREPOQA achieves consistent 4-10% improvements over state-of-the-art baselines across multiple LLM models, with gains concentrated on Completeness and Reasoning dimensions critical for multi-hop repository understanding. Ablation studies identify MCTS as the primary performance driver, while exploration analysis reveals that a middle-node budget provides optimal quality–efficiency trade-offs. Our work shows the effectiveness of systematic tree search for repository-level reasoning and provides a foundation for building more capable software engineering assistants

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

# A APPENDIX

## A.1 USE OF LARGE LANGUAGE MODELS

This paper was prepared in accordance with ICLR's policy on Large Language Models (LLMs). In this work, LLMs were used to aid or polish writing, for example to improve grammar, clarity, and phrasing throughout the manuscript. LLMs were not used for retrieval and discovery (e.g., finding related work) or for research ideation.

## A.2 ACTION SPACE AND ARGUMENTS

Table 4 summarizes the available actions and their required arguments used by our system. The action space is designed to be both comprehensive and concise, providing the agent with a powerful set of tools for repository exploration. Actions can be categorized into name-based retrieval ('FindClass', 'FindFunction', 'FindCodeSnippet'), which allows for precise lookups when a specific identifier is known, and semantic retrieval ('SemanticSearch'), which enables conceptual searches based on natural language queries. The 'ViewCode' action allows the agent to inspect specific code regions returned by searches, while the 'Finish' action signals the termination of the search process. This structured action space guides the agent's reasoning and ensures that its interactions with the codebase are grounded and efficient.

Table 4: Action options in DEEPREPOQA.

| Action | Arguments | Embedding? |
|---|---|---|
| **FindClass** | class-name, file-pattern | No |
| **FindFunction** | function-name, class-name, file-pattern | No |
| **FindCodeSnippet** | code-snippet, file-pattern | No |
| **ViewCode** | code-span-list, file-pattern | No |
| **SemanticSearch** | query, category, file-pattern | Yes |
| **Finish** | answer, finish-reason | No |

## A.3 ALGORITHM OF DEEPREPOQA

We provide the pseudo code of DEEPREPOQA in Algorithm 1. The algorithm operates in three main phases. In the initialization phase, a search tree is created with the user's question and repository context as the root. The core of the algorithm is the iterative search phase, where for a fixed number of iterations, it traverses the tree using the UCT policy (selection), expands the tree with new nodes (expansion), simulates the outcome of actions to estimate their value (simulation), and updates the values of nodes along the path (back-propagation). The process can terminate early if a "Finish" action is selected. Finally, in the result extraction phase, the best trajectory from the completed search paths is identified, and the final answer is synthesized from it.

---

**Algorithm 1:** DEEPREPOQA Algorithm

**Input:** User question $Q$, Repository context $R$, Max iterations $N$
**Output:** Final answer $A$

1  search_tree ← MCTSTree($Q$, $R$);
2  iteration_count ← 0;
3  max_iterations ← $N$;
4  **while** *iteration_count < max_iterations* **do**
5      iteration_count ← iteration_count + 1;
6      node ← Select(search_tree.root);      // Select promising node with highest UCT
        score, except those that have 3+ children
7      new_node ← Expand(node);
8      situational_report ← Perceive(new_node, new_node.parent, new_node.siblings);
9      new_node.action ← PlanNextAction($Q$, situational_report);
10     raw_result ← ExecuteAction(new_node.action);
11     new_node.observation ← FilterEvidence($Q$, raw_result);
12     value ← Evaluate(new_node);
13     Backpropagate(new_node, value);
14     **if** *new_node.action.type = "Finish"* **then**
15         **break**;
16     **end**
17  **end**
18  finished_nodes ← GetFinishedNodes(search_tree);
19  $T$ ← GetBestTrajectory(finished_nodes);
20  $A$ ← ExtractAnswer($T$);
21  **return** $A$

---

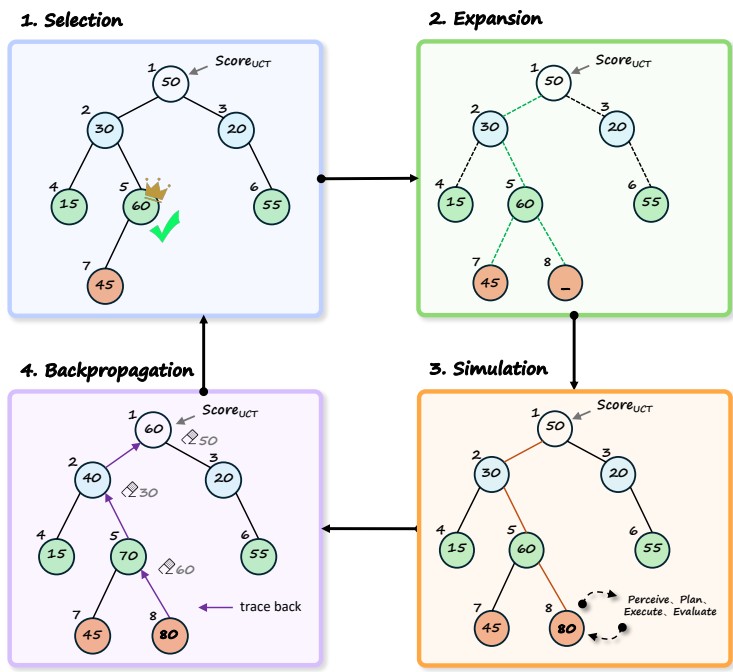

Figure 5: Illustration of our MCTS workflow. Four phases—selection (UCT), expansion, simulation, and back-propagation—guide exploration; values/prior from LLM feedback shape UCT scores, preventing loops and focusing on high-utility branches.

### A.4  DETAILS OF OUR MCTS WORKFLOW

The MCTS engine guides the repository exploration of DEEPREPOQA through a systematic four-phase process: selection, expansion, simulation, and back-propagation. This structure is designed to navigate the complex search space of repository-level reasoning efficiently. An illustration of our MCTS workflow is provided in Figure 5.

- **Selection:** Starting from the root, the algorithm recursively selects child nodes based on the Upper Confidence Bound for Trees (UCT) policy, which balances exploration of new paths and exploitation of known valuable paths. For example, node5, having the highest UCT score and being eligible for expansion, is selected.

- **Expansion:** If the selected leaf node is not fully expanded, a new child node is added to the tree. Here, node5 is expanded to generate node8.

- **Simulation:** Instead of a full rollout, our approach uses a single-step simulation involving our four agents. The *Perception Agent* first generates a situational report to inform the expansion and avoid redundant explorations. Then the *Planning Agent* proposes an action, the *Execution Agent* performs it and gathers evidence, and the *Evaluation Agent* assigns a value to the action's outcome. This value directly updates the Q-value of the state-action pair. After the simulation, node8 received a value of 80 points.

- **Back-propagation:** The value obtained from the simulation is propagated back up the tree from the new node to the root, updating the visit counts and Q-values of all nodes along the path. This ensures that more promising branches are explored more frequently in subsequent iterations. In Back-propagation stage, node5's value increased from 60 to 70, node2's value increased from 30 to 40. The process continues until reaching the termination.

This iterative process allows DEEPREPOQA to dynamically focus its search on the most promising reasoning paths, leading to more effective and efficient question answering.

Table 5: Results by question type. DEEPREPOQA is strongest on Why/What and remains competitive on How/Where, indicating robust architectural and definition-level grounding with precise localization.

| Question Type | GPT-4o | DeepSeek-V3.1 | Claude-Sonnet-3-7 | Average |
|---|---|---|---|---|
| What | 40.21 | 44.57 | 44.97 | 43.25 |
| Architecture exploration | 37.73 | 43.34 | 45.52 | 42.20 |
| Concept / Definition | 43.45 | 46.11 | 46.00 | 45.19 |
| Dependency tracing | 39.44 | 44.26 | 43.40 | 42.37 |
| Why | 40.55 | 46.48 | 45.25 | 44.09 |
| Design rationale | 41.31 | 47.37 | 44.31 | 44.99 |
| Purpose Exploration | 41.70 | 46.87 | 45.44 | 44.67 |
| Performance | 38.65 | 45.20 | 46.00 | 43.28 |
| Where | 38.64 | 44.75 | 46.26 | 43.22 |
| Data / Control-flow | 36.65 | 44.13 | 45.67 | 42.82 |
| Feature Location | 37.27 | 44.17 | 45.92 | 42.45 |
| Identifier Location | 42.00 | 45.94 | 47.18 | 45.71 |
| How | 35.70 | 42.82 | 44.07 | 40.86 |
| System Design | 34.90 | 42.28 | 44.15 | 40.44 |
| Algorithm Implementation | 37.48 | 43.44 | 45.00 | 41.97 |
| API / Framework Support | 34.71 | 42.73 | 43.06 | 40.17 |

## A.5 RESULTS ACROSS DIFFERENT QUESTION TYPES

Table 5 shows the performance of DEEPREPOQA across different question types. The results indicate that DEEPREPOQA performs strongly across all major question categories, with "Why" and "What" questions achieving the highest average scores. This suggests the model is particularly effective at understanding design rationale and fetching definitions or concepts. "Where" questions, which often require precise locations of features or identifiers, also see strong performance. "How" questions, which typically demand a deeper understanding of system design or algorithmic implementation, are the most challenging, yet our method still achieves competitive scores. This detailed breakdown highlights the versatility of our approach and points to areas for future improvement in handling complex procedural queries.

## A.6 PROMPTS USED IN DEEPREPOQA

This section details the key prompt templates that steer the behavior of the LLM agents within the DEEPREPOQA framework. These prompts are engineered to provide clear instructions and context for different modules, including the main ReAct-style agent, the perceiving agent for MCTS guidance, the agent responsible for identifying relevant code spans from search results, the value estimation agent, and the LLM-as-a-Judge. Each prompt is designed to elicit the desired reasoning process and output format, ensuring that the agents operate effectively within their designated roles.

---

**Prompt Template for Planning**

```
You are an autonomous AI assistant with superior skills in answering questions of a
repository.
You need to answer the question exactly based on the information you can get from the
available functions.
As you're working autonomously, you cannot communicate with the user but must rely on
information you can get from the available functions.

# Action and ReAct Guidelines

1. Analysis First
   - Review all previous actions and their observations
   - Understand what has been done and what information you have

2. Document Your Thoughts
   - ALWAYS write your reasoning in <thoughts> tags before any action
   - Explain what you learned from previous observations
```

```
        - Justify why you're choosing the next action
        - Describe what you expect to learn/achieve

3. Single Action Execution
    - Run ONLY ONE action at a time
    - Choose from the available functions
    - Never try to execute multiple actions at once

4. Wait and Observe
    - After executing an action, STOP
    - Wait for the observation (result) to be returned
    - Do not plan or execute any further actions until you receive the observation

# Workflow Overview

1. Understand the Question
    * Review the Question: Carefully read the question provided in <task>
    * Identify Needed Information: Analyze the question to determine what parts of the
    codebase you need to understand
    * Plan Your Investigation: Determine what components, functions, or classes you need
    to explore to find a complete answer

2. Locate Relevant Code
    * Primary Method - Search Functions:
        - FindClass - Search for class definitions by class name
        - FindFunction - Search for function definitions by function name
        - FindCodeSnippet - Search for specific code patterns or text
        - SemanticSearch - Search code by semantic meaning or natural language description
    * Secondary Method - ViewCode: Only use when you need to see:
        - Additional context not returned by searches
        - Specific line ranges from search results
        - Code referenced in other parts of the codebase

3. Gather Complete Information
    - Continue searching and viewing code until you have all necessary information
    - Investigate all relevant parts of the codebase
    - Look for related functionality that might be important

4. Analyze and Formulate Answer
    - Review all gathered information
    - Organize findings to create a complete and accurate answer
    - Ensure the answer is based on actual code, not assumptions

5. Complete Task
    - Use Finish you have sufficient information to provide a complete and accurate answer
    - Reference specific parts of the code in your final answer
    - Make sure the answer addresses all aspects of the question

# Important Guidelines

- Focus on the Specific Question
  - Answer exactly as asked, based on the code in the repository
  - Provide complete and accurate information
  - Do not assume code you haven't seen

- Code Context and Information
  - Base answers only on code you can see through searches and ViewCode
  - Use ViewCode to examine more code if needed
  - Reference specific code sections for clarity

- Task Completion
  - Finish only after gathering sufficient information
  - Cite specific evidence from the code
  - Ensure all relevant code has been explored

- State Management
  - Keep detailed records of viewed code and actions taken
  - Check history before performing a new action to avoid repetition
  - Use gathered information to inform next steps

# Additional Notes

- Think Step by Step
  - Document reasoning and thought process in <thoughts>
  - Build upon previous steps without unnecessary repetition

- Never Guess
  - Do not guess line numbers or code content
  - Use ViewCode to examine code when needed
```

```
# Examples

Here are some examples of how to use the available actions:

**Example 1:
Task: I need to see the implementation of the DatabaseManager class to understand how
it handles transactions
    <thoughts>To examine how the DatabaseManager class handles transactions, we need to
    locate its implementation in the codebase.</thoughts>
    {"tool": "FindClass", "file_pattern": null, "class_name": "DatabaseManager"}

**Example 2:
Task: Find the calculate_interest function in our financial module to review its logic
    <thoughts>To review the logic of the calculate_interest function, we need to locate
    its implementation in the financial module.</thoughts>
    {"tool": "FindFunction", "file_pattern": "financial/**/*.py", "function_name":
    "calculate_interest", "class_name": null}

*Note: This is a condensed version for display purposes. The actual prompt contains
complete examples for all action tools.*
```

## Prompt Template for Perceiving and Generating Feedback

```
You are a feedback agent that guides an AI assistant's next action.

---
##  CRITICAL: ACTION AGENT LIMITATIONS

The action agent receiving your feedback:

- CANNOT see the search tree
- Has NO CONTEXT about node relationships
- Only knows about actions in its direct trajectory
- Cannot understand references to nodes without proper context
- Is at a new node that has NO ACTION YET { it needs your guidance for what to do next

---
##  REQUIRED FEEDBACK STRUCTURE

### 1. CURRENT NODE CONTEXT
You must start by describing:

- **Position in tree:** '"You are at Node X, which is [position relative to root]"'
- **Current state:** '"Your node is currently empty and awaiting your first action"'
- **Parent context:** '"Your parent node (Node Y) [describe what parent did]"'
- **Relationship to solutions:** '"There are [N] terminal nodes in [relationship]
branches"'

> **Note:** The current node is ALWAYS empty and awaiting its first action { never
describe it as having done something already.

--
##  CORRECT EXAMPLES

**Current Node Context Example:**
"You are at Node 8, which is your first action from the root.
Your node is currently empty and awaiting your first action.
Your parent (Node 1) performed a FindCodeSnippet action that didn't add new context.
There are three terminal nodes in parallel branches (Nodes 7, 9, and 14) that have
reached finished states with different approaches."

---
##  INCORRECT EXAMPLES { DO NOT USE

- '"Node 8 is empty and expandable"'
- '"The current node needs to explore improvements"'
- '"We should validate the existing solution"'
- Any description implying the current node has already taken an action

---
##  INPUT STRUCTURE

1. **Tree Visualization:** ASCII representation showing:
    - Node IDs and relationships
    - Action types at each node
```

```
    - Key outcomes and observations

2. **Original Task:** The problem to solve

3. **Message History:** Chain of executed actions leading to current state

4. **Tree Structure:**
   - **Parent Node:** Your current starting point, the last successfully executed
   action
   - **Current Node:** Your branch from the parent, waiting for your next action
   - **Sibling Nodes:** Other independent solution attempts branching from the same
   parent
      (These are from different trajectories and have not happened in your current path)

5. **Alternative Node Information:**
   - Proposed actions and parameters
   - Outcomes (from separate, independent trajectories)
   - Warning flags for previously attempted approaches

---
##  YOUR TASK

1. **Analyze the situation:**
   - Start with current node context (position, state, parent, solutions)
   - Consider sibling attempts (alternative universes)
   - Learn from outcomes to avoid repeating unsuccessful approaches
   - Contextualize feedback based on tree structure
   - Always explain node relationships and attempts
   - Inform about alternative approaches tried (files, tests, git diffs)

2. **Suggest next action:**
   - Clear, actionable guidance focusing on code query actions
   - Based on lessons from other attempts
   - Avoid repeating failed approaches

3. **Optionally suggest node to expand:**
   - Explain why this node is promising
   - Leave as null if no strong preference

> Remember: Focus on code query actions that help the agent better understand the
codebase. Always provide proper context since the agent cannot see the tree.

**Important:** Please provide the answer in JSON format.
```

## Prompt Template for Identifying Useful Code Spans

```
You are an autonomous AI assistant tasked with identifying relevant code in a codebase.
Your goal is to select key code sections from the search results that are most relevant
to the search request.

---

## Context

The previous messages will contain:

1. A search request from an AI assistant
2. Search results containing various code sections with their line numbers

---

## Your Task

### 1. Understand the Search Request
- Analyze the previous search request to understand what elements are being looked for
- Identify key elements such as **functions, variables, classes, or patterns** that are
relevant

### 2. Evaluate Search Results
- Examine each code section in the search results for alignment with the search request
- Assess the **relevance and importance** of each code section
- Consider the **complete context** of code sections

### 3. Respond with the Identify Action
- Select and respond with the **code sections** that best match the search request
- Provide your **analysis in the thoughts field**
```

```
    - List the relevant **file paths** with **start and end line numbers** in the
    `identified_spans` field
```

## Prompt Template for Node Evaluation

```
Your role is to evaluate the **last executed action** of the search tree that our AI
agents are traversing, to help determine the best trajectory to solve a programming
issue. The agent is responsible for identifying and modifying the correct file(s)
in response to the problem statement.

**Important:** While line numbers may be referenced in the initial problem description,
they can shift as changes are made to the file. Focus on whether the agent is modifying
the correct logical parts of the code, rather than strictly matching the initially
mentioned line numbers. What matters is that the right section of code is being
modified, even if its current line number differs from what was originally specified.

## Task

At this stage, the agent is still working on the solution. Your task is twofold:

1. **Evaluation**: Assess whether the change done by the **last executed action** is
appropriate for addressing the problem and whether the agent is on the right path to
resolving the issue. Verify that the correct sections of code are being modified,
regardless of their current line numbers.

2. **Alternative Feedback**: Independently of your evaluation, provide guidance for
an alternative problem-solving branch. This ensures parallel exploration of different
solution paths.

## Evaluation Criteria

- **Exploratory Actions:** Initial searches and information-gathering steps are
essential and should not be heavily penalized if they don't yield immediate results.
- **Appropriateness of Action:** Is the action logical given the agent's current
knowledge and early stage of problem-solving?
- **Query Relevance:** Is the search query or parameters well-defined and likely to
find relevant code?
- **Search Scope Appropriateness:** Do the file patterns and class/function names
narrow down the search effectively?
- **Relevance of Search Results:** Are the search results directly related to the
problem and useful for progress?
- **Size of Search Results:** Is the code context appropriately sized (not too large
or too small)?
- **Category Appropriateness:** Does the category (implementation or test) align with
the search intent?

## Reward Scale

Assign an integer between 0 and 100:

- 90-100: Excellent action; parameters are well-defined and results are highly
relevant.
- 75-89: Good action; parameters are reasonable and results are relevant.
- 25-74: Action has issues or yields few/no relevant results.
- 0-24: Counterproductive; results are irrelevant or excessively large.

## Feedback Structure

- **Explanation:** Provide detailed reasoning for your decision, focusing on the **last
executed action**, its relation to previous actions, and its impact.
- **Feedback to Alternative Branch:** Suggest conceptual alternative approaches without
actual code, avoiding actions that would replicate previous outcomes.
- **Reward:** Assign a single integer between 0 and 100 based on confidence in
correctness and likelihood of solving the issue.

## Available Actions

### FindClass
Use this when you know the exact class name to find.

- Finding class implementations: `class_name="UserRepository"`
- Locating test classes: `class_name="TestUserAuthentication"`
- Finding base classes: `class_name="BaseController"`
- Classes in specific modules: `class_name="Config", file_pattern="src/config/*.py"`

### FindFunction
Use when you know the exact function or method name.
```

```
- Finding test cases: `function_name="test_user_login"`
- Locating implementations: `function_name="process_payment"`
- All methods with a name: `function_name="validate"`
- Specific class method: `function_name="save", class_name="UserRepository"`

### FindCodeSnippet
Use when you know the exact code snippet.

- Finds constants: `code_snippet="MAX_RETRIES = 3"`
- Finds decorators: `code_snippet="@retry(max_attempts=3)"`
- Finds imports: `code_snippet="from datetime import datetime"`
- Configuration patterns: `code_snippet="DEBUG = os.getenv('DEBUG', False)"`

> Note: If you don't know the exact code, use **SemanticSearch**.

### SemanticSearch
Use when you don't know exact names but want related functionality.

- Functionality by description: `query="code that handles password hashing"`
- Related test cases: `query="tests for user registration", category="test"`
- Implementations: `query="database connection pooling", category="implementation"`
- Error handling patterns: `query="error handling for API requests"`

> Flexible for unknown names, discovering related code, or exploring features.

### ViewCode
View the code in a file or a specific span.

### Finish
Indicate that the generated code answer is accurate and complete for the user's query.
```

## Prompt for LLM-as-Judge

```
You are a professional evaluator. Please rate the candidate answer against the
reference answer based on five criteria.

Evaluation Criteria and Scoring Guidelines (each scored 1 to 10):
1.Correctness:
    10 - Completely correct; core points and details are accurate with no ambiguity.
    8-9 - Mostly correct; only minor details are slightly inaccurate or loosely
          expressed.
    6-7 - Partially correct; some errors or omissions, but main points are generally
          accurate.
    4-5 - Several errors or ambiguities that affect understanding of the core
          information.
    2-3 - Many errors; misleading or fails to convey key information.
    1 - Serious errors; completely wrong or misleading.

2. Completeness:
    10 - Covers all key points from the reference answer without omission.
    8-9 - Covers most key points; only minor non-critical information missing.
    6-7 - Missing several key points; content is somewhat incomplete.
    4-5 - Important information largely missing; content is one-sided.
    2-3 - Covers very little relevant information; seriously incomplete.
    1 - Covers almost no relevant information; completely incomplete.

3. Relevance:
    10 - Content fully focused on the question topic; no irrelevant information.
    8-9 - Mostly focused; only minor irrelevant or peripheral information.
    6-7 - Generally on topic; some off-topic content but still relevant overall.
    4-5 - Topic not sufficiently focused; contains considerable off-topic content.
    2-3 - Content deviates from topic; includes excessive irrelevant information.
    1 - Majority of content irrelevant to the question.

4. Clarity:
    10 - Fluent language; clear and precise expression; very easy to understand.
    8-9 - Mostly fluent; clear expression with minor unclear points.
    6-7 - Generally clear; some expressions slightly unclear or not concise.
    4-5 - Expression somewhat awkward; some ambiguity or lack of fluency.
    2-3 - Language obscure; sentences are not smooth; hinders understanding.
    1 - Expression confusing; very difficult to understand.

5. Reasoning:
    10 - Reasoning is clear, logical, and well-structured; argumentation is excellent.
    8-9 - Reasoning is clear and logical; well-structured with solid argumentation.
    6-7 - Reasoning generally reasonable; mostly clear logic; minor jumps.
    4-5 - Reasoning is average; some logical jumps or organization issues.
```

```
        2-3 - Reasoning unclear; lacks logical order; difficult to follow.
        1 - No clear reasoning; logic is chaotic.

    INPUT:
        Question:{question}
        Reference Answer:{reference}
        Candidate Answer:{candidate}

    OUTPUT:
        Please output ONLY a JSON object with 5 integer fields in the range [1,10],
        corresponding to the evaluation scores:
            {{
                "correctness":  <1-10>,
                "completeness": <1-10>,
                "relevance":    <1-10>,
                "clarity":      <1-10>,
                "reasoning":    <1-10>
            }}

    REQUIREMENT:
        No explanation, no extra text, no formatting other than valid JSON"""
```

## A.7 SUPPLEMENTARY EXPERIMENTS

We conducted supplementary experiments on three additional repositories (`conan`, `reflex` and `streamlink` from `swe-bench-live`) from the SWE-QA Bench dataset, using GPT-4o as the model. We also added `OpenHands` and `SWE-agent` as new baselines. For each method, we evaluated both the QA performance, using the same metrics as in the main experiments, and the computational cost, including latency and token usage, to assess efficiency.

### A.7.1 EXPERIMENTAL SETUP

- **OpenHands_QA**: uses `openhands-sdk` v1.1.0 and `openhands-tools` v1.1.0, with the agent set to `get_default_agent` (which includes tools such as `terminal`, `file_editor`, `task_tracker`, `finish`, `think`, etc.).

- **SWE-agent_QA**: based on `SWE-agent` v1.0, retaining all tools and the core agent logic; only minor modifications were made to the entry/exit points to adapt it to QA tasks.

- **Iteration limit**: For DEEPREPOQA, SWE-QA-Agent, OpenHands_QA and SWE-agent_QA, `max_iteration` is set to 10. When the limit is reached, the responses are generated using `message_history` (DEEPREPOQA only uses the adopted trajectory, not all node messages).

### A.7.2 RESULTS

**Performance**  Table 6 reports the supplementary QA results on three additional SWE-QA Bench repositories. Our method, DEEPREPOQA, consistently achieves the highest scores across most evaluation metrics, including Correctness, Completeness, and Reasoning, demonstrating its superior factual accuracy, coverage, and logical soundness. Compared to the original baselines, DEEPREPOQA shows substantial improvements—for example, it outperforms Prompt-Direct by over 18 points in total score. Even against commercial tools and newly introduced baselines such as OpenHands_QA and SWE-agent_QA, DEEPREPOQA maintains competitive performance, highlighting its effectiveness in generating high-quality answers across diverse repository contexts.

**Computational Efficiency**  Table 7 summarizes the computational efficiency of all methods. Our method, DEEPREPOQA, achieves competitive latency while maintaining high-quality outputs, demonstrating a good balance between performance and computational cost. Although some simpler baselines such as Prompt-Direct exhibit lower latency and fewer tokens, they generate lower-quality answers. Compared to commercial tools and new baselines, DEEPREPOQA maintains moderate response time with substantially fewer input tokens than OpenHands, indicating efficient utilization of computational resources without compromising answer quality. Overall, these results highlight that DEEPREPOQA achieves excellent speed, efficient token usage, and superior answer quality across diverse repositories.

Table 6: Supplementary QA results on additional SWE-QA Bench repositories.

| Method | Evaluation Metrics | | | | | Total |
|--------|-------------|--------------|---------|-----------|-----------|-------|
| | Correctness | Completeness | Clarity | Relevance | Reasoning | |
| DEEPREPOQA | 8.19 | 7.95 | 8.82 | 9.43 | 8.39 | 42.78 |
| *Original Baseline* | | | | | | |
| Prompt-Direct | 4.01 | 2.36 | 7.03 | 7.77 | 3.11 | 24.28 |
| RAG_Sliding_window | 6.42 | 4.66 | 7.38 | 8.88 | 5.16 | 32.50 |
| RAG_Func_chunk | 6.56 | 4.69 | 7.36 | 8.90 | 5.79 | 33.30 |
| SWE-QA-Agent | 7.09 | 5.76 | 7.96 | 9.27 | 6.82 | 36.90 |
| *Commercial Tools* | | | | | | |
| Cursor-agent-2025.09.04 | 8.55 | 7.77 | 8.73 | 9.74 | 8.52 | 43.31 |
| *New Baselines* | | | | | | |
| OpenHands_QA | 7.80 | 7.05 | 8.50 | 9.28 | 7.84 | 40.47 |
| SWE-agent_QA | 8.23 | 7.60 | 8.72 | 9.49 | 8.35 | 42.39 |

Table 7: Latency and token usage for supplementary experiments

| Method | Metrics | | |
|--------|-------------|--------------|---------------|
| | Latency (s) | Input Tokens | Output Tokens |
| DEEPREPOQA | 70.15 | 92,070.28 | 4,490.02 |
| *Original Baseline* | | | |
| Prompt-Direct | 0.84 | 103.51 | 40.71 |
| RAG_Sliding_window | 5.14 | 3,887.53 | 441.56 |
| RAG_Func_chunk | 5.26 | 3,557.34 | 433.24 |
| SWE-QA-Agent | 14.17 | 58,054.82 | 349.24 |
| *Commercial Tools* | | | |
| Cursor-agent-2025.09.04 | 60.22 | 129,458.33 | 1,635.19 |
| *New Baselines* | | | |
| SWE-agent_QA | 88.72 | 111,444.09 | 2,024.50 |
| OpenHands | 18.88 | 286,447.48 | 4,774.88 |

