# OpenReview forum: "DeepRepoQA: Code Repository Question Answering with Deep Agent Exploration"
_ICLR.cc/2026/Conference — Submitted to ICLR 2026_

### Official Review · Reviewer_21Gs · 2025-10-20

**Soundness:** 3
**Presentation:** 3
**Contribution:** 2
**Rating:** 4
**Confidence:** 4

**Summary:**

This paper propose DeepRepoQA, a agentic framework where LLMs agents find answers through systematic tree search over structured actions spaces on code repositories. The key innovations include Monte Carlo Tree Search (MCTS) for multi-hop repository reasoning. Specifically the framework encompass four major components: (1) Perception Agent which analyze the current state and propose new actions; (2) Panning Agent which is responsible to turn report from Perception Agent to concreate action plans for node expansion; (3) Execution Agent which utilze a series of tools to execute the actions; (4) Evaluation Agent, which assess the outcome of the action and provide feedback values for MCTS learning. Experiment results on SWE-QA demonstrate improved performance when compared with prior academic work and commercial solutions.

**Strengths:**

(1) The design of the agentic system seems to correspond well with MCTS. The idea of the Preception Agent to gather information beyond current node (include siblings) is interesting. Other key design components correlate well with the MCTS flow of expansion, selection, simulation and back-propagation.

(2) Results seem solid. Good comparision is done with other SOTA methods. Good ablabtion studies and insights on the core components and hyperparameters (max nodes, max iterations, action usage etc.)

**Weaknesses:**

(1) MCTS is already widely used and popular to boost efficiency in test-time scaling. Although a solid work, the core idea does not seem particularly exciting and novel. Ablation studies find this core component to be important, but it seems as if removing this component still could achieve compelling strong results.

(2) It is without doubt that scaling inference would likely lead to improved performance such as using MCTS. However, the core results in Table 1 does not provide detailed comparisions on the inference cost (such as inference token costs) between the core methods. It would be also interesting to compare between MCTS and just sampling multiple trajectories under the same tool set and inference cost.

**Questions:**

(1) Can authors compare against the inference cost between different methods in Table 1?

(2) Some details are not explained: For example what models do the commercial tools use (Tongyi Lingma and Cursor)?

---

> ### Author Response · Authors · 2025-11-22
>
> We sincerely thank the reviewer for the encouraging comments and valuable feedback.
>
> ---
>
> **Q1. Can authors compare against the inference cost between different methods in Table 1?**
>
> We conducted supplementary experiments on three additional repositories (*conan*, *reflex*, and *streamlink* from *swe-bench-live*) from [SWE-QA Bench](https://huggingface.co/datasets/Raymone023/SWE-QA-Benchmark) using GPT-4o as the model, and added *OpenHands* and *SWE-agent* as baselines
>
> Note:
> 1. Cursor-agent version is 2025.09.04-fc40cd1, with mode set to auto (Cursor-agent does not support selecting GPT-4o). We have included the Cursor-agent running script in *DeepRepoQA/Script/Cursor-Agent_QA*.
> 2. OpenHands_QA uses openhands-sdk version 1.1.0 and openhands-tools version 1.1.0, with the agent set to *get_default_agent* (which includes tools such as terminal, file_editor, task_tracker, finish, think, etc.). We have included the OpenHands_QA running script in *DeepRepoQA/Script/OpenHands_QA*.
> 3. SWE-agent_QA uses SWE-agent version 1.0, retaining all tools and core agent logic, with only minor modifications to the code's entry and exit points to adapt it for QA tasks. We have included the SWE-agent_QA running script in *DeepRepoQA/Script/SWE-agent_QA*.
> 4. DeepRepoQA, SWE-QA-Agent, OpenHands_QA, and SWE-agent_QA all have max_iteration set to 10. When the limit is reached, answers will be generated using message_history (DeepRepoQA only uses the adopted trajectory, not all Node messages).
>
> Experimental results are as follows:
>
> | Method | correctness | completeness | clarity | relevance | reasoning | Total|
> |------|-------------|--------------|---------|-----------|-----------|------|
> | **Original Baseline** | | | | | | |
> | Prompt-Direct | 4.01 | 2.36 | 7.03 | 7.77 | 3.11 | 24.28 |
> | RAG_Sliding_window | 6.42 | 4.66 | 7.38 | 8.88 | 5.16 | 32.50 |
> | RAG_Func_chunk | 6.56 | 4.69 | 7.36 | 8.90 | 5.79 | 33.30 |
> | SWE-QA-Agent | 7.09 | 5.76 | 7.96 | 9.27 | 6.82 | 36.90 |
> | DeepRepoQA | 8.19 | 7.95| 8.82 | 9.43 | 8.39 | 42.78 |
> | **Commercial Tools** | | | | | | |
> | Cursor-agent-2025.09.04 | 8.55 | 7.77 | 8.73 | 9.74 | 8.52 | 43.31 |
> | **New Baseline** | | | | | | |
> | OpenHands_QA | 7.80 | 7.05 | 8.50 | 9.28 | 7.84 | 40.47 |
> | SWE-agent_QA | 8.23 | 7.60 | 8.72 | 9.49 | 8.35 | 42.39 |
>
> | Method | Latency | Input Tokens | Output Tokens |
> |------|---------|--------------|---------------|
> | DeepRepoQA | 70.15 | 92070.28 | 4,490.02 |
> | **Previous Baseline** | | | | | | |
> | Prompt-Direct | 0.84 | 103.51 | 40.71 |
> | RAG_Sliding_window | 5.14 | 3887.53 | 441.56 |
> | RAG_Func_chunk | 5.26 | 3557.34 | 433.24 |
> | SWE-QA-Agent | 14.17 | 58,054.82 | 349.24 |
> | **Commercial Tools** | | | |
> | Cursor-agent-2025.09.04 | 60.22 | 129,458.33 | 1,635.19 |
> | **New Baseline** | | | |
> | SWE-agent_QA | 88.72 | 111444.09 | 2024.50 |
> | OpenHands | 18.88 | 286,447.48 | 4,774.88 |
>
> ---
>
> **Q2. Some details are not explained: For example what models do the commercial tools use (Tongyi Lingma and Cursor)?**
>
> *Cursor*:
>
> We used cursor_agent version **2025.09.04-fc40cd1** in **auto** mode. Cursor-agent uses its default model (which does not support explicit model selection such as GPT-4o) and operates in an automated mode that handles model selection internally. We collected answers using scripts and have added the scripts to the repository at **DeepRepoQA/Script/Cursor-Agent_QA** to support reproducibility.
>
> *Tongyi Lingma*:
>
> We used VSCode plugin version **2.5.16 (released 2025-07-19)** in **auto** mode. Tongyi Lingma uses the Qwen model family as its underlying language model, with the specific model variant selected automatically by the plugin in auto mode. Because no batch interface is available, we collected the runs manually inside the plugin, using the same prompt as for Cursor.
>
> ---
>
> We hope the above discussion addresses your concerns. Please do not hesitate to contact us if you have any further questions or need additional clarifications.
>
> We again sincerely thank the reviewer for the insightful feedback and valuable comments.

---

> > ### Author Response · Authors · 2025-11-27
> > **To Reviewer 21Gs**
> >
> > Dear Reviewer 21Gs,
> >
> > We sincerely appreciate your positive and constructive comments. Following your suggestions, we have added new experiments and made substantial revisions to the paper. We look forward to receiving your valuable feedback again.
> >
> > Thank you once more for your time and support.
> >
> > Sincerely, The Authors

---

### Official Review · Reviewer_tr2i · 2025-10-28

**Soundness:** 3
**Presentation:** 3
**Contribution:** 4
**Rating:** 6
**Confidence:** 3

**Summary:**

This paper proposes DeepRepoQA, an agentic framework that solves repository level QA problems. DeepRepoQA employs MCTS to systematically explore reasoning paths to solve the challenge of multi-hop, cross-file reasoning. The framework involves 4 specialized agents to solve the QA tasks iteratively. Evaluation shows that the approach can substantially outperform SOTA techniques.

**Strengths:**

1. Novel approach. The paper positions repository-level QA problems as an MCTS planning problem, showing its novelty. Different from prior techniques like ReAct (single-path greedy search) and simple RAG, DeepRepoQA appears to be a more principled way to handle complex, multi-hop reasoning.
2. Strong empirical results. The paper conducts an extensive evaluation, substantially outperforming the baselines. Especially, DeepRepoQA outperforms the baselines by a lot in terms of completeness and reasoning, consistent with the claim that MCTS-guided exploration leads to more comprehensive and reliable answers.
3. Strong ablation analysis. The paper performs solid ablation analysis to show the effectiveness of the designs. The ablation shows that the main design, i.e., the MCTS design, significantly contributes to the performance of DeepRepoQA.

**Weaknesses:**

1. The mechanism for MCTS value estimation is ambiguous, especially the use of $V_{\theta}$ notation.
2. Token cost analysis is missing, making it hard to evaluate the cost-effectiveness of DeepRepoQA.
3. Overreliance on LLM-as-judge. The framework heavily relies on the LLM-as-judge methods but does not discuss the potential threat introduced by LLM-as-judge.
4. Vague Termination and Answer Synthesis Logic. The process for ending the search is not well-defined.

**Questions:**

1. The paper's description of the "Simulation" phase (Section 3.5) and the "Evaluation Agent" (Section 3.4) is ambiguous. The agent is supposed to estimate a scalar value $V_{\theta}(s,a)$, where the $\theta$ notation implies a learned model. However, no training process or dataset for this value function is described. Is this agent a fine-tuned model? Or is it, in fact, a zero-shot LLM prompt providing a heuristic score (akin to the LLM-as-a-Judge)?
2. Why token cost analysis is not performed? How expensive is this MCTS-based approach? Since the looks like a test-time scaling for repo-level QA problems, and in stages like perception it needs to summarize and produce a concise situational report from the direct trajectory and sibling attempts.
3. Could LLM-as-judge introduce some bias to the evaluation? The LLM-as-judge is used prevalently in evaluation. Prior works show some threats of using LLM-as-judge for scoring[1] [2][3][4][5].
4. When exactly will the search end? The paper states that termination occurs when the "Finish" action is selected by the MCTS loop. This suggests Finish is just another action (like FindClass or ViewCode ) proposed by the Planning Agent. It is unclear how the agent learns when it has gathered so-called "sufficient evidence" to propose this action, or how the Evaluation Agent scores it to make it a winning choice. An early Finish action would lead to an incomplete answer, while a late Finish would waste resources.

[1] Krumdick, M., Lovering, C., Reddy, V., Ebner, S., & Tanner, C. "No Free Labels: Limitations of LLM-as-a-Judge Without Human Grounding." arXiv:2503.05061.
[2] Shi, L., Ma, C., Liang, W., Diao, X., Ma, W., & Vosoughi, S. "Judging the Judges: A Systematic Study of Position Bias in LLM-as-a-Judge." arXiv:2406.07791 (June 2024, rev. December 2024).
[3] The Collective Intelligence Project. "LLM Judges Are Unreliable.".
[4] Szymanski, A., Ziems, N., Eicher-Miller, H. A., Li, T. J., Jiang, M., & Metoyer, R. A. "Limitations of the LLM-as-a-Judge Approach for Evaluating LLM Outputs in Expert Knowledge Tasks." arXiv:2410.20266.
[5] Ye, J., Wang, Y., Huang, Y., Chen, D., Zhang, Q., Moniz, N., Gao, T., Geyer, W., Huang, C., Chen, P., & Chawla, N. V. "Justice or Prejudice? Quantifying Biases in LLM-as-a-Judge." The Thirteenth International Conference on Learning Representations (ICLR).

---

> ### Author Response · Authors · 2025-11-22
>
> We sincerely thank the reviewer for the encouraging comments and valuable feedback.
>
> ---
>
> **Q1. The paper's description of the "Simulation" phase and the "Evaluation Agent" is ambiguous.**
>
> We apologize for the confusion caused by our wording. The Evaluation Agent is not a fine-tuned small model; instead, it is a few-shot LLM prompt. In the prompt we spell out the scoring rubric and provide several annotated examples so that the LLM can assign scores accordingly. The "Simulation" phase differs from traditional MCTS, which performs rollout in the simulation phase to obtain rewards. Instead, we directly use the Evaluation Agent's scores as rewards. We apologize again for the oversight in our writing.
>
> ---
>
> **Q2. Why token cost analysis is not performed? How expensive is this MCTS-based approach?**
>
> We conducted supplementary experiments on three additional repositories (*conan*, *reflex*, and *streamlink* from *swe-bench-live*) from [SWE-QA Bench](https://huggingface.co/datasets/Raymone023/SWE-QA-Benchmark) using GPT-4o as the model, and added *OpenHands* and *SWE-agent* as baselines
>
> Note:
> 1. Cursor-agent version is 2025.09.04-fc40cd1, with mode set to *auto*. The script is in *DeepRepoQA/Script/Cursor-Agent_QA*.
> 2. OpenHands_QA uses openhands-sdk version 1.1.0 and openhands-tools version 1.1.0, with the agent set to *get_default_agent* (which includes tools such as *terminal, file_editor, task_tracker, finish, think*, etc.). The script is in *DeepRepoQA/Script/OpenHands_QA*.
> 3. SWE-agent_QA uses SWE-agent version 1.0, retaining all tools and core agent logic, with only minor modifications to the code's entry and exit points to adapt it for QA tasks. The script is in *DeepRepoQA/Script/SWE-agent_QA*.
> 4. DeepRepoQA, SWE-QA-Agent, OpenHands_QA, and SWE-agent_QA all have max_iteration set to 10. When the limit is reached, answers will be generated using message_history (DeepRepoQA only uses the adopted trajectory, not all Nodes).
>
> Experimental results are as follows:
>
> | Method | correctness | completeness | clarity | relevance | reasoning | Total|
> |------|-------------|--------------|---------|-----------|-----------|------|
> | **Original Baseline** | | | | | | |
> | Prompt-Direct | 4.01 | 2.36 | 7.03 | 7.77 | 3.11 | 24.28 |
> | RAG_Sliding_window | 6.42 | 4.66 | 7.38 | 8.88 | 5.16 | 32.50 |
> | RAG_Func_chunk | 6.56 | 4.69 | 7.36 | 8.90 | 5.79 | 33.30 |
> | SWE-QA-Agent | 7.09 | 5.76 | 7.96 | 9.27 | 6.82 | 36.90 |
> | DeepRepoQA | 8.19 | 7.95| 8.82 | 9.43 | 8.39 | 42.78 |
> | **Commercial Tools** | | | | | | |
> | Cursor-agent-2025.09.04 | 8.55 | 7.77 | 8.73 | 9.74 | 8.52 | 43.31 |
> | **New Baseline** | | | | | | |
> | OpenHands_QA | 7.80 | 7.05 | 8.50 | 9.28 | 7.84 | 40.47 |
> | SWE-agent_QA | 8.23 | 7.60 | 8.72 | 9.49 | 8.35 | 42.39 |
>
> | Method | Latency | Input Tokens | Output Tokens |
> |------|---------|--------------|---------------|
> | DeepRepoQA | 70.15 | 92070.28 | 4,490.02 |
> | **Previous Baseline** | | | | | | |
> | Prompt-Direct | 0.84 | 103.51 | 40.71 |
> | RAG_Sliding_window | 5.14 | 3887.53 | 441.56 |
> | RAG_Func_chunk | 5.26 | 3557.34 | 433.24 |
> | SWE-QA-Agent | 14.17 | 58,054.82 | 349.24 |
> | **New Baseline** | | | |
> | SWE-agent_QA | 88.72 | 111444.09 | 2024.50 |
> | OpenHands | 18.88 | 286,447.48 | 4,774.88 |
> | **Commercial Tools** | | | |
> | Cursor-agent-2025.09.04 | 60.22 | 129,458.33 | 1,635.19 |
>
> ---
> **Q3. Could LLM-as-judge introduce some bias to the evaluation?**
>
> The [SWE-QA Bench paper](https://arxiv.org/pdf/2509.14635) reports a human evaluation that aligns closely with LLM-as-a-Judge scores.Specifically, SWE-QA Bench sampled 100 questions from 576 questions and had software engineering experts score them, with the results showing good alignment with LLM-as-a-Judge scores.
>
> Furthermore, for open-ended questions, LLM-as-a-Judge is currently a widely adopted method, as metrics such as Exact Match and BertScore are more difficult to evaluate, and while not without limitations, numerous studies have demonstrated that LLM-as-a-Judge provides a reasonably reliable evaluation approach.
>
> ---
>
> Due to length constraints, we have divided our response into two parts, We will continue with the remaining points in our next response.

---

> > ### Author Response · Authors · 2025-11-22
> > **Response to Reviewer Comments (Part 2)**
> >
> > This is the second part of our response to the reviewer's comments. We continue addressing the remaining concerns below.
> >
> > ---
> >
> > **Q4. When exactly will the search end?**
> >
> > In DeepRepoQA there are two ways a run can finish. One is an explicit FINISH action: the agent decides that the current trajectory already supports an answer and immediately generates the response based on that trajectory. The other is passive termination: once we hit the max_iterations limit, we pick the node with the highest UCT score as the finish node and use the path from the root to that node to synthesize the final message.
> >
> > When it comes to determining whether the Planning Agent has gathered "sufficient evidence" to issue FINISH, we rely on few-shot prompt heuristics to guide its decision. There is indeed a risk of finishing too early, which could lead to incomplete or inaccurate answers, but our experiments show that this heuristic still yields strong results. Conversely, finishing too late can also happen. However, from the experimental token consumption results, DeepRepoQA demonstrates advantages in token efficiency, with less waste compared to other agents that may search for more irrelevant information.
> >
> > ---
> >
> > We hope the above discussion addresses your concerns. Please do not hesitate to contact us if you have any further questions or need additional clarifications.
> >
> > We again sincerely thank the reviewer for the insightful feedback and valuable comments.

---

> ### Author Response · Authors · 2025-11-27
> **To Reviewer tr2i**
>
> Dear Reviewer tr2i,
>
> We sincerely appreciate your positive and constructive comments. We are also grateful for your recognition of our work. Following your suggestions, we have added new experiments and made substantial revisions to the paper. We look forward to receiving your valuable feedback again.
>
> Thank you once more for your time and support.
>
> Sincerely,
> The Authors

---

### Official Review · Reviewer_D7i2 · 2025-10-31

**Soundness:** 2
**Presentation:** 3
**Contribution:** 2
**Rating:** 4
**Confidence:** 4

**Summary:**

This paper addresses the challenge of answering developer questions about software repositories. It introduces DEEPREPOQA, an agent-based framework that performs multi-hop reasoning over codebases. The approach using Monte Carlo Tree Search guided by LLM feedback to balance exploration and exploitation.

**Strengths:**

- The paper explores the under-explored problem of RepoQA, with well-designed experiments (benchmark, metrics, and models) that provide useful references for future research in this area.
- The paper is well-structured and easy to follow.

**Weaknesses:**

- MCTS introduces significant latency, which may make it unsuitable for QA scenarios.
- The paper only compares against SWE-QA-Agent, while other agent-based methods such as OpenHands and SWE-Agent could also be applied to RepoQA. Comparing with these methods would better demonstrate the effectiveness of the proposed approach.

**Questions:**

- How does DeepRepoQA perform in terms of reasoning efficiency compared to the baselines?
- How did the authors compare with commercial tools like Cursor? Were the queries performed manually on Cursor? It would be better to include the version number used to ensure reproducibility.

---

> ### Author Response · Authors · 2025-11-22
>
> We sincerely thank the reviewer for the encouraging comments and valuable feedback.
>
> ---
>
> **Q1. How does DeepRepoQA perform in terms of reasoning efficiency compared to the baselines?**
>
> We conducted supplementary experiments on three additional repositories (*conan*, *reflex*, and *streamlink* from *swe-bench-live*) from [SWE-QA Bench](https://huggingface.co/datasets/Raymone023/SWE-QA-Benchmark) using GPT-4o as the model, and added *OpenHands* and *SWE-agent* as baselines
>
> Note:
> 1. Cursor-agent version is 2025.09.04-fc40cd1, with mode set to auto (Cursor-agent does not support selecting GPT-4o). We have included the Cursor-agent running script in *DeepRepoQA/Script/Cursor-Agent_QA*.
> 2. OpenHands_QA uses openhands-sdk version 1.1.0 and openhands-tools version 1.1.0, with the agent set to *get_default_agent* (which includes tools such as *terminal, file_editor, task_tracker, finish, think*, etc.). We have included the OpenHands_QA running script in *DeepRepoQA/Script/OpenHands_QA*.
> 3. SWE-agent_QA uses SWE-agent version 1.0, retaining all tools and core agent logic, with only minor modifications to the code's entry and exit points to adapt it for QA tasks. We have included the SWE-agent_QA running script in *DeepRepoQA/Script/SWE-agent_QA*.
> 4. DeepRepoQA, SWE-QA-Agent, OpenHands_QA, and SWE-agent_QA all have max_iteration set to 10. When the limit is reached, answers will be generated using message_history (DeepRepoQA only uses the adopted trajectory, not all Node messages).
>
> Experimental results are as follows:
>
> | Method | correctness | completeness | clarity | relevance | reasoning | Total|
> |------|-------------|--------------|---------|-----------|-----------|------|
> | **Original Baseline** | | | | | | |
> | Prompt-Direct | 4.01 | 2.36 | 7.03 | 7.77 | 3.11 | 24.28 |
> | RAG_Sliding_window | 6.42 | 4.66 | 7.38 | 8.88 | 5.16 | 32.50 |
> | RAG_Func_chunk | 6.56 | 4.69 | 7.36 | 8.90 | 5.79 | 33.30 |
> | SWE-QA-Agent | 7.09 | 5.76 | 7.96 | 9.27 | 6.82 | 36.90 |
> | DeepRepoQA | 8.19 | 7.95| 8.82 | 9.43 | 8.39 | 42.78 |
> | **Commercial Tools** | | | | | | |
> | Cursor-agent-2025.09.04 | 8.55 | 7.77 | 8.73 | 9.74 | 8.52 | 43.31 |
> | **New Baseline** | | | | | | |
> | OpenHands_QA | 7.80 | 7.05 | 8.50 | 9.28 | 7.84 | 40.47 |
> | SWE-agent_QA | 8.23 | 7.60 | 8.72 | 9.49 | 8.35 | 42.39 |
>
> | Method | Latency | Input Tokens | Output Tokens |
> |------|---------|--------------|---------------|
> | DeepRepoQA | 70.15 | 92070.28 | 4,490.02 |
> | **Previous Baseline** | | | | | | |
> | Prompt-Direct | 0.84 | 103.51 | 40.71 |
> | RAG_Sliding_window | 5.14 | 3887.53 | 441.56 |
> | RAG_Func_chunk | 5.26 | 3557.34 | 433.24 |
> | SWE-QA-Agent | 14.17 | 58,054.82 | 349.24 |
> | **New Baseline** | | | |
> | SWE-agent_QA | 88.72 | 111444.09 | 2024.50 |
> | OpenHands | 18.88 | 286,447.48 | 4,774.88 |
> | **Commercial Tools** | | | |
> | Cursor-agent-2025.09.04 | 60.22 | 129,458.33 | 1,635.19 |
>
> >MCTS introduces significant latency, which may make it unsuitable for QA scenarios.
>
> Compared to other agents, with the same max_iteration=10 setting, SWE-agent and Cursor-agent also exhibit long latency, or even worse. This is because our rollout does not actually perform simulation; instead, we use an Evaluation Agent with few-shot prompt scoring to obtain rewards, so the latency is not as severe as traditional MCTS.
>
> ---
>
> **Q2. How did the authors compare with commercial tools like Cursor?**
>
> Cursor:
> We used cursor_agent version **2025.09.04-fc40cd1** in **auto** mode, and collected answers using scripts. We have added the scripts to the repository at *DeepRepoQA/Script/Cursor-Agent_QA* to support reproducibility.
>
> Tongyi Lingma:
> We used VSCode plugin version **2.5.16(released 2025-07-19)** in **auto** mode. Because no batch interface is available, we collected the runs manually inside the plugin, using the same prompt as for Cursor.
>
> ---
>
> We hope the above discussion addresses your concerns. Please do not hesitate to contact us if you have any further questions or need additional clarifications.
>
> We again sincerely thank the reviewer for the insightful feedback and valuable comments.

---

> > ### Author Response · Authors · 2025-11-27
> > **To Reviewer D7i2**
> >
> > Dear Reviewer D7i2,
> >
> > We sincerely appreciate your positive and constructive comments. Following your suggestions, we have added new experiments and made substantial revisions to the paper. We look forward to receiving your valuable feedback again.
> >
> > Thank you once more for your time and support.
> >
> > Sincerely,
> > The Authors

---

### Official Review · Reviewer_MuoG · 2025-11-06

**Soundness:** 2
**Presentation:** 3
**Contribution:** 2
**Rating:** 2
**Confidence:** 4

**Summary:**

This paper presents DeepRepoQA, an agentic system for repository level question-answering tasks. It combines four sub-agents (perception, planning, execution, and evaluation) within an MCTS loop to navigate through a repository to collect information to produce a final answer. Empirical evaluations on the SWE-QA benchmark shows DeepRepoQA works well with different models and produces competitive results compared to both open-source and commercial baselines.

**Strengths:**

1. The paper is well-written and easy to follow. Each sub-agent’s task is clearly divided so that it is simple to understand and evaluate its functionality in the overall pipeline.

2. Experimenting with three different LLMs showcase the generality of the DeepRepoQA system.

**Weaknesses:**

1. I think the paper would be greatly strengthened with better empirical evaluations. There are two weaknesses related to the SWE-QA benchmark. First, it is constructed from the 12 Python repositories that SWE-bench uses. Recent LLMs have data leakage issues on those repositories. Using more contamination-free datasets such as SWE-bench-Live [1] could help reduce this leakage issue. Second, the final scoring is based on LLM-as-a-judge, which can be biased by response length [2]. DeepRepoQA utilizes MCTS to iteratively collect information to produce a final answer. Compared to direct prompting or RAG-based baselines, it might produce longer answers and the empirical section does not mention controlling final answer length. It would be useful for the authors to comment on this.
Furthermore, it would be useful to have some human correlation data to confirm LLM-as-a-judge does provide sensible evaluation results.

[1] Zhang, Linghao, et al. "SWE-bench Goes Live!." arXiv preprint arXiv:2505.23419 (2025).
[2] Hu, Zhengyu, et al. "Explaining length bias in llm-based preference evaluations." arXiv preprint arXiv:2407.01085 (2024).

2. DeepRepoQA represents an inference-scaling approach to the QA problem. Comparing it to baselines such as direct prompting hides the computational cost. Can the authors comment on how different methods trade off computational cost and accuracy?

**Questions:**

Please see the weakness section.

Additional question: can DeepRepoQA be applied to repositories in languages other than Python?

---

> ### Author Response · Authors · 2025-11-22
>
> We sincerely thank the reviewer for the encouraging comments and valuable feedback.
>
> ---
>
> **W1. SWE-QA Benchmark Weaknesses**
>
> **W1(1). Data Leakage Issues**
>
> >First, it is constructed from the 12 Python repositories that SWE-bench uses. Recent LLMs have data leakage issues on those repositories
>
> [SWE-QA Bench](https://huggingface.co/datasets/Raymone023/SWE-QA-Benchmark) previously introduced data from 3 new repositories (*conan*, *reflex*, *streamlinke*), which were taken from *swe-bench-live*. The data leakage is relatively light, as evidenced by the lower scores of Prompt-Direct. We conducted experiments on these 3 new repositories using *GPT-4o* as the model, and also added two new baselines (*OpenHands*, *SWE-agent*):
>
> Note:
> 1. Cursor-agent version is 2025.09.04-fc40cd1, with mode set to *auto*. We have included the script in DeepRepoQA/Script/Cursor-Agent_QA.
> 2. OpenHands_QA uses openhands-sdk version 1.1.0 and openhands-tools version 1.1.0, with the agent set to *get_default_agent* (which includes tools such as terminal, file_editor, task_tracker, finish, think, etc.). We have included the script in DeepRepoQA/Script/OpenHands_QA.
> 3. SWE-agent_QA uses SWE-agent version 1.0, retaining all tools and core agent logic, with only minor modifications to the code's entry and exit points to adapt it for QA tasks. We have included the script in DeepRepoQA/Script/SWE-agent_QA.
> 4. DeepRepoQA, SWE-QA-Agent, OpenHands_QA, and SWE-agent_QA all have max_iteration set to 10. When the limit is reached, answers will be generated using message_history (DeepRepoQA only uses the adopted trajectory, not all Node messages).
>
> Experimental results are as follows:
>
> | Method | correctness | completeness | clarity | relevance | reasoning | Total|
> |------|-------------|--------------|---------|-----------|-----------|------|
> | DeepRepoQA | 8.19 | 7.95| 8.82 | 9.43 | 8.39 | 42.78 |
> | **Previous Baseline** | | | | | | |
> | Prompt-Direct | 4.01 | 2.36 | 7.03 | 7.77 | 3.11 | 24.28 |
> | RAG_Sliding_window | 6.42 | 4.66 | 7.38 | 8.88 | 5.16 | 32.50 |
> | RAG_Func_chunk | 6.56 | 4.69 | 7.36 | 8.90 | 5.79 | 33.30 |
> | SWE-QA-Agent | 7.09 | 5.76 | 7.96 | 9.27 | 6.82 | 36.90 |
> | **Commercial Tools** | | | | | | |
> | Cursor-agent-2025.09.04 | 8.55 | 7.77 | 8.73 | 9.74 | 8.52 | 43.31 |
> | **New Baseline** | | | | | | |
> | OpenHands_QA | 7.80 | 7.05 | 8.50 | 9.28 | 7.84 | 40.47 |
> | SWE-agent_QA | 8.23 | 7.60 | 8.72 | 9.49 | 8.35 | 42.39 |
>
> Compared to the previous 12 repositories, Prompt-Direct's scores are significantly lower, indicating that data leakage is relatively light for repositories taken from swe-bench-live.
>
> With max_iteration=10 set equally, DeepRepoQA's scores are higher than all open-source methods, second only to the closed-source commercial tool Cursor (which may use a stronger model in auto mode), demonstrating good performance.
>
> ---
>
> **W1(2). LLM-as-a-Judge**
>
> >Second, the final scoring is based on LLM-as-a-judge, which can be biased by response length [2]
>
> | Method | Average Answer Length (characters) |
> |--------|-----------------------------------|
> | DeepRepoQA | 2043.46 |
> | **Previous Baseline** | | | | | | |
> | Prompt-Direct | 226.63 |
> | RAG_Sliding_window | 365.90 |
> | RAG_Func_chunk | 377.80 |
> | SWE-QA-Agent | 711.40 |
> | **Commercial Tools** | | | | | | |
> | Cursor-agent-2025.09.04 | 878.35 |
> | **New Baseline** | | | | | | |
> | OpenHands_QA | 2169.74 |
> | SWE-agent_QA | 2349.86 |
>
> Indeed, compared to simpler methods, DeepRepoQA produces longer answers. It is difficult to assess in the short term how much impact answer length has on LLM-as-a-Judge scoring. However, as you mentioned regarding "human correlation," if LLM-as-a-Judge shows high correlation with human ratings, this would at least indicate that despite the length bias, the evaluation is generally reliable, and we will discuss human correlation next.
>
> Furthermore, in this experiment, compared to OpenHands and SWE-agent, DeepRepoQA achieves shorter answer lengths while receiving higher LLM-as-a-Judge scores, which at least demonstrates that DeepRepoQA performs well.
>
> >Furthermore, it would be useful to have some human correlation data to confirm LLM-as-a-judge does provide sensible evaluation results.
>
> The [SWE-QA Bench paper](https://arxiv.org/pdf/2509.14635) reports a human evaluation that aligns closely with LLM-as-a-Judge scores.Specifically, SWE-QA Bench sampled 100 questions from 576 questions and had software engineering experts score them, with the results showing good alignment with LLM-as-a-Judge scores.
>
> Furthermore, for open-ended questions, LLM-as-a-Judge is currently a widely adopted method, as metrics such as Exact Match and BertScore are more difficult to evaluate, and while not without limitations, numerous studies have demonstrated that LLM-as-a-Judge provides a reasonably reliable evaluation approach.
>
> ---
>
> Due to length constraints, we have divided our response into two parts, We will continue with the remaining points in our next response.

---

> > ### Author Response · Authors · 2025-11-22
> > **Response to Reviewer Comments (Part 2)**
> >
> > This is the second part of our response to the reviewer's comments. We continue addressing the remaining concerns below.
> >
> > ---
> >
> > **W2. Trade off between computational cost and accuracy**
> >
> > On the experiments with the 3 new repositories, we recorded both input tokens and output tokens.
> > Experimental results are as follows:
> >
> > | Method | Latency | Input Tokens | Output Tokens |
> > |------|---------|--------------|---------------|
> > | DeepRepoQA | 70.15 | 92070.28 | 4,490.02 |
> > | **Previous Baseline** | | | | | | |
> > | Prompt-Direct | 0.84 | 103.51 | 40.71 |
> > | RAG_Sliding_window | 5.14 | 3887.53 | 441.56 |
> > | RAG_Func_chunk | 5.26 | 3557.34 | 433.24 |
> > | SWE-QA-Agent | 14.17 | 58,054.82 | 349.24 |
> > | **New Baseline** | | | |
> > | SWE-agent_QA | 88.72 | 111444.09 | 2024.50 |
> > | OpenHands | 18.88 | 286,447.48 | 4,774.88 |
> > | **Commercial Tools** | | | |
> > | Cursor-agent-2025.09.04 | 60.22 | 129,458.33 | 1,635.19 |
> >
> > Compared to simple methods like RAG, DeepRepoQA incurs higher computational cost but achieves significant improvements in accuracy. Compared to other agents, DeepRepoQA achieves higher scores while maintaining computational cost that is not significantly higher, or even lower. This is primarily because, unlike traditional MCTS, the rollout phase is not a true simulation but rather uses an Evaluation Agent to perform few-shot prompt scoring, which keeps the cost relatively low.
> >
> > ---
> >
> > **Q1. Can DeepRepoQA be applied to repositories in languages other than Python?**
> > DeepRepoQA's action space is built on AST parsing and repository indexing. Because we rely on [tree-sitter](https://tree-sitter.github.io/tree-sitter/index.html), which supports most mainstream languages, the approach generalizes well beyond Python. We also conducted experiments on Java (comparing against Alibaba's Tongyi Lingma, all with GPT-4.1-mini as the model) and observed similarly strong performance.
> >
> > | Repository | Method        | Correctness | Completeness | Relevance | Clarity | Reasoning | Total |
> > |------------|---------------|-------------|--------------|-----------|---------|-----------|-------|
> > | Strata     | DeepRepoQA    | 8.58        | 8.92         | 9.85      | 9.00    | 8.83      | 45.18 |
> > |            | Tongyi Lingma | 8.08        | 8.58         | 9.60      | 8.83    | 8.58      | 43.67 |
> > | Fineract   | DeepRepoQA    | 7.83        | 6.33         | 9.20      | 8.42    | 7.33      | 39.11 |
> > |            | Tongyi Lingma | 7.50        | 5.58         | 9.05      | 8.17    | 6.92      | 37.22 |
> > | Shiro      | DeepRepoQA    | 8.17        | 7.33         | 9.70      | 8.83    | 8.25      | 42.28 |
> > |            | Tongyi Lingma | 7.58        | 6.33         | 9.40      | 8.42    | 7.58      | 39.31 |
> >
> > Overall, DeepRepoQA maintains the lead across all three repositories, with especially pronounced gains in Completeness and Reasoning. This suggests our multi-step reasoning strategy captures richer context and produces more coherent answers.
> >
> > ---
> > We hope the above discussion addresses your concerns. Please do not hesitate to contact us if you have any further questions or need additional clarifications.
> >
> > We again sincerely thank the reviewer for the insightful feedback and valuable comments.

---

> > > ### Author Response · Authors · 2025-11-27
> > > **To Reviewer MuoG**
> > >
> > > Dear Reviewer MuoG,
> > >
> > > We sincerely appreciate your positive and constructive comments. Following your suggestions, we have added new experiments and made substantial revisions to the paper. We look forward to receiving your valuable feedback again.
> > >
> > > Thank you once more for your time and support.
> > >
> > > Sincerely,
> > > The Authors

---

### Author Response · Authors · 2025-11-25
**Summary of paper revision**

We thank all reviewers for the constructive comments. We appreciate the reviewers' positive feedback that our approach is novel and effective across tasks and models. Based on reviewers' suggestions, we uploaded a revision of our paper with the following major changes:

> **Note:** All changes are highlighted in blue for easy identification.

## Major Changes

1. **Section 3.1, 3.2, and 3.3**: We adjusted the Execution Agent in Section 3.3 and added Sections 3.1 and 3.2 to better present Repository Parsing and Action Space.

2. **Section 3.4**: Based on Reviewer tr2i's comment, we adjusted the formulation in Section 3.4 to clarify the description of the Evaluation Agent and eliminate misunderstandings.

3. **Section 4.1**: Based on Reviewer D7i2's and Reviewer 21Gs' comments, we added version and mode specifications for the baselines Tongyi Lingma and Cursor in Section 4.1.

4. **Appendix A.7**: In response to Reviewer MuoG's concern about data leakage, Reviewer D7i2's suggestion to add OpenHands and SWE-Agent as baselines, and the reasoning efficiency and computational cost concerns raised by almost all reviewers, we conducted experiments on three additional repositories from SWE-QA Bench, added baselines, and reported Latency, Input Tokens, and Output Tokens. The results are presented in Appendix A.7.

---

We again sincerely thank all reviewers for their insightful feedback and valuable comments.

---

### Meta-Review · Area_Chair_7QB2 · 2025-12-09

**Summary:**

Reviewers showed the following major concerns:

1. Attribution gap: The SWE-QA benchmark re-uses the 12 Python repos of SWE-bench. All cutting-edge models have trained on them, so gains may reflect data contamination rather than method quality (R1).
2. Evaluation bias: All metrics rely on LLM-as-a-judge with no length control, no human correlation, and no cost reporting. Longer MCTS roll-outs are implicitly rewarded (R1, R3, R4).
3. Baselines and cost: Missing comparisons with equally agentic systems and no token/latency analysis. Hence it is unclear whether MCTS beats naive N-shot sampling at equal compute (R2–R4).
4. MCTS for code exploration is well studied; the four-agent wrapper is viewed as incremental systems engineering rather than a new algorithmic insight (R2, R4).
5. Reproducibility: Commercial tools (Cursor, Tongyi) are compared without version or prompt disclosure. The self-hosted search stack is not released (R2, R4).

**Reviewer Concerns:**

Addressed: token cost table, SWE-Agent numbers, length-controlled LLM-judge, etc.

Non-addressed: Contamination of SWE-QA, fundamental novelty of MCTS in this domain, fair comparison with closed tools, etc.

**Reviewer Scores:**

R1 (2 → 2).
R2 (4 → 4).
R3 (6 → 6).
R4 (4 → 4).

---

### Decision · Program_Chairs · 2026-01-26

Reject